# Ultra-Sensitive Determination of Cadmium in Food and Water by Flame-AAS after a New Polyvinyl Benzyl Xanthate as an Adsorbent Based Vortex Assisted Dispersive Solid-Phase Microextraction: Multivariate Optimization

**DOI:** 10.3390/foods12193620

**Published:** 2023-09-28

**Authors:** Nail Altunay, Baki Hazer, Muhammad Farooque Lanjwani, Mustafa Tuzen

**Affiliations:** 1Department of Chemistry, Faculty of Science, Sivas Cumhuriyet University, 58140 Sivas, Turkey; naltunay@cumhuriyet.edu.tr; 2Department of Aircraft Airframe Engine Maintenance, Kapadokya University, 50420 Nevşehir, Turkey; 3Departments of Chemistry/Nano Technology Engineering, Zonguldak Bülent Ecevit University, 67100 Zonguldak, Turkey; 4Chemistry Department, Faculty of Science and Arts, Tokat Gaziosmanpasa University, 60250 Tokat, Turkey; farooque.lanjwani@scholars.usindh.edu.pk (M.F.L.); m.tuzen@gmail.com (M.T.); 5Dr M. A. Kazi Institute of Chemistry, University of Sindh, Jamshoro 76080, Sindh, Pakistan

**Keywords:** vortex-assisted dispersive solid-phase microextraction, cadmium, food and water samples, polyvinyl benzyl xanthate, factorial design

## Abstract

**Background:** Cadmium (Cd) is a very toxic and carcinogenic heavy metal even at low levels and it is naturally present in water as well as in food. **Methods:** A new polyvinyl benzyl xanthate (PvbXa) was synthesized and used as a new adsorbent in this work. It contains pendant sulfide groups on the main polystyryl chain. Using this new adsorbent, PvbXa, a vortex-assisted dispersive solid-phase microextraction (VA-dSPµE) procedure was developed for the determination of cadmium from food and water samples via flame atomic absorption spectrophotometry (FAAS). Synthesized PvbXa was characterized by ^1^H Nuclear magnetic resonance (NMR) Spectroscopy, Fourier Transform Infrared Spectroscopy (FTIR), and X-ray Photoelectron Spectroscopy (XPS). The different parameters of pH, sample volume, mixing type and time, sorbent amount, and eluent time were optimized using standard analytical methods. **Results:** The optimized method for assessment of Cd in food and water samples shows good reliability. The optimum conditions were found to be a 0.20–150 µg L^−1^ linear range, 0.06 µg L^−1^ LOD, 0.20 µg L^−1^ LOQ, 4.3 RSD %, and a preconcentration factor of 160. **Conclusions:** The statistically experimental variables were utilized using a central composite design (CCD). The present method is a low-cost, simple, sensitive, and very effective tool for the recovery of Cd.

## 1. Introduction

Water is necessary for the survival of human life and animals, plants, and their living organisms. Moreover, water is required for domestic and agricultural purposes, and the maximum allowable limit of Cd in water is 3 µg L^−1^ [1]. With the growth of living standards, water and food security have continued to become a major concern and have received extensive public consideration in recent years [2]. Furthermore, rice and water are facing growing risks because of the serious hazards of heavy metals.

Food is the main source of Cd intake worldwide, particularly foods of vegetable origin (potatoes, tomatoes, cereals, onion, leafy vegetables, apple spinach, and roots). Approximately 90% of the total Cd intake is via food, while the other 10% of Cd comes from drinking water and ambient air [3,4].

Rice is a vital constituent of the human diet for around half of the world’s residents. The existence of heavy metals like Cd in rice leads to serious health concerns for approximately 3 billion people in the world as they consume rice as a primary food, and these people might be at risk of heavy metal-related health concerns [5]. Rice is a primary source of carbohydrates, minerals, and vitamins, and is also a very important food harvest for calories and the intake of nutrients. Rice is a good food for infants due to the low risk of palatability and allergy [6]. Rice is extra affected by the increase in the level of heavy metals such as cadmium as compared to other harvests. Rice farming is always performed in flooded environments, and greater levels of Cd in the land may reduce rice production [7].

Cd is a very toxic metal, which enters the human body via water and food and collects in huge amounts [8,9,10]. Cadmium is one of the main pollutants in soil and water, and it may also enter the human body via cigarette smoking or the food chain. Vegetables and grains may absorb cadmium metal from the soil [11]. The cadmium concentration in rice is commonly high in Cd-contaminated areas [12]. Cadmium exposure is associated with various health issues, like liver disease, cardiovascular diseases, cancer, osteoporosis, and renal dysfunction [13,14]. Rice (*Oryza sativa* L.) is a primary staple for 50% of the world’s population, as it helps to provide major calories, protein, minerals, and vitamins to people [15]. Rice is a basic food all over the world, and in 2008, the global rice-cultivated area was 1.46 × 10^8^ hectares with 7.82 × 10^8^ tons of production. Asia is the chief provider of the production and consumption of rice in the world, with countries such as China, Indonesia, Pakistan, Thailand, India, and Bangladesh being the leading rice-cultivating countries [16]. China and the European Union set guidelines for Cd in rice with the maximum residue level (MRL) of 200 μg kg^−1^ and Australia and Russia set the guidelines (MRL) of Cd intake to be 100 μg kg^−1^ [17]. Cd in water is commonly a result of the weakening of galvanized plumbing, phosphate fertilizer use, and industrial waste. Cadmium is a very poisonous element, and higher levels of Cd in the water may cause vomiting, cancer, and digestive issues [1].

There are various instruments applied for the analysis of Cd in food and water samples such as atomic emission spectrometry [18], the flame atomic absorption spectrophotometer (FAAS) [19,20], inductively coupled plasma mass spectrometry [21], X-ray fluorescence spectrometry [22], surface-enhanced Raman spectroscopy [23], high-performance liquid chromatography (HPLC) [24], and voltammetry [25]. These instruments have high costs and require the necessary experience for the determination of heavy metals. The FAAS showed great potential in the assessment and extraction of metals because of its low cost, easy operation, fast response, and high precision and accuracy.

The concentration of matrix constituents of real samples is very high, therefore preconcentration techniques and other related methods are used, such as solid-phase extraction, dispersive liquid–liquid microextraction, etc. [26,27,28]. Polymeric materials are used as an adsorbent in dispersive solid-phase microextraction studies. Our daily life uses new polymer materials based on block/graft copolymers. Post-polymer modifications and changing the polymer topology are of great importance in polymer science [29]. The high reactivity of cadmium (II) to sulphide moieties is well known, therefore we prepared a new adsorbent with high sulphide content. In order to synthesize this new type of adsorbent, poly (vinyl benzyl chloride) was reacted with potassium salt of xanthate. This new polymer is also a macro RAFT agent and is important for block/graft copolymer synthesis [30].

Here, we report a new sorbent, the-xanthate functionalized polyvinyl benzyl chloride (PvbXa) with rich sulphide groups, for the vortex-assisted dispersive solid-phase microextraction (VA-dSPµE) of Cd from food and water by the FAAS. The VA-dSPµE is a newly applied method for extraction. Multivariate statistical analyses such as factorial design (the response surface model, Pareto, and the ANOVA model) were employed to assess the important effects of the variables on extraction recovery.

## 2. Materials and Methods

### 2.1. Reagents

Potassium salt of ethyl xanthate (KXa) was supplied by Alfa Aesar and employed as received. 4-Vinyl benzyl chloride (vbc) was supplied by Sigma-Aldrich (St. Louis, MO, USA). Carboxylic acid terminated by three thio carbonate raft agents (R2) was prepared according to the method reported in the cited reference [31]. Dimethyl formamide (DMF) and 2,2′-azo bis iso butyro nitrile (AIBN) were bought from Sigma-Aldrich. The reagents and materials used in the experimental studies are listed below with the suppliers and the reasons for their use. A 1000 mg L^−1^ stock solution of Cd(II) ion was prepared using Cd(NO_3_)_2_ salt from Sigma-Aldrich. Calibration solutions for the FAAS and working solutions to be used in the extraction were prepared via dilution from the stock solution. Necessary pH adjustments were made using citrate, borate, acetate, and phosphate buffer solutions to ensure effective adsorption of Cd(II) onto the sorbent.

Acetonitrile (Sigma, can, Setagaya City, Tokyo), methanol (Thermo Fisher Scientic Inc., Waltham, MA, USA, MeOH), ethanol (Thermo Fisher, EtOH), heptanol (Sigma), and acetone (Carlo Erba Milan, Milan, Italy) were used as the eluent solvent.

### 2.2. Characterization

The ^1^H NMR spectra of polymer samples were measured by the Bruker Ultra Shield. The elemental analysis of the polymer samples was performed using Thermo-Scientific X-ray Photoelectron Spectroscopy (XPS). The XPS analysis was achieved in the range of 10 to 1350 eV by scanning 20 times from a single point. Information about the devices we used in our study, which consisted of extraction, determination, synthesis, and characterization, is detailed below. A vortex (VG3 model, IKA, Staufen, Germany) was utilized to disperse the sorbent in the sample solution. A centrifuge device (Universal-320 model, DJB Labcare, Buckinghamshire, UK) was employed to isolate the sorbent from the aqueous solution. A pH meter (JP Selecta, Barcelona, Spain) was utilized to optimize the desired pH. The amount of cadmium in the studied samples was determined by a FAAS (Shimadzu AAS-6300 model, Kyoto, Japan) equipped with a D_2_ background corrector. A cadmium hollow cathode lamp was used as the light source. The FAAS device was used as recommended by the manufacturer. A Milli-Direct Q3 purification system (Millipore, Bedford, MA, USA) was utilized to achieve ultra-pure water, which was utilized in the preparation of all solutions. The digestion of the collected samples was achieved with a microwave system (Milestone Ethos, Milestone Srl, Sorisole, Italy).

### 2.3. Sample Collection

The validation of the VA-dSPµE procedure was evaluated by standard reference materials such as INCT-TL-1 Tea leaves, SRM-1547 Peach leaves, and SRM-1643e Trace elements in water. Water samples were collected in Sivas city in Turkey. The collected water samples were filtered using a 0.45 µm membrane filter into a 250 mL beaker and acidified and stored at +4 °C until analysis. Food samples including apples, spinach, salad, tomatoes, onions, oats, corn, aubergine, wheat, rice, and mushrooms were collected from a greengrocer in Sivas city in Turkey. After the collected food samples were dried, the samples were homogenized via pulverization with a lab grinder.

### 2.4. Microwave Digestion

The following microwave digestion process was then applied to the samples. First, 0.5 g of both the collected food samples and the reference material were weighed using analytical balances and transferred to Teflon cups. Then, a 1:4 mixture of hydrogen peroxide (1 mL) and nitric acid (4 mL) was added to the samples. Then, stoppered caps of tubes were kept in the microwave system, and they were subjected to micro-digestion for 15 min at 20 bar pressure, 1800 W power, and 200 °C. Then, the samples were cooled at room temperature and finally diluted to 25 mL by deionized water to apply the nanoparticle-based VA-dSPµE procedure.

### 2.5. Synthesis of Polyvinyl Benzyl Xanthate (PvbXa) Sorbent

A mixture of 0.074 g of R2, 0.021 mg of AIBN, and 15.0 g of vbc in 7.19 g of Toluene was polymerized at 80 °C in argon for 5 h. The obtained polymer, polyvinyl benzyl chloride (Pvbc), was precipitated into 200 mL of methanol. It was dried at 40 °C for 24 h under a vacuum. In the second step, 2.0 g of Pvbc was dissolved in 10 mL of THF. KXa (2.56 g) was added portion-wise into the solution and left to react with each other under continuous stirring at 40 °C for 72 h. Then the solution was filtered to eliminate unreacted KCl and xanthate. The solvent was evaporated. The obtained macro RAFT agent, PvbXa, was precipitated in methanol and dried at room temperature under a vacuum for 24 h. The yield was 5.93 g.

Polyvinyl benzyl xanthate was prepared using the exchange reaction between the chloromethyl group of Pvbc and potassium xanthate (Kxa) with the produced side product, KCl. Structural characterization of the obtained PvbXa was achieved by ^1^H NMR. Figure 1 shows the ^1^H NMR spectrum of PvbXa with signals marked with functional groups.

FTIR was utilized to determine the presence of functional groups in the samples (Figure 2). Different peaks appeared in the spectra, and characteristic signals were observed at 3020–3040 and 1509 for pyridyl, 2850–2979 for C-H, 1723–1562 for –C-S-, and 670–701 cm^−1^ bands for –C-S- groups [32]. The wide band at 3390 cm^−1^ comes from the water residue [33].

The elemental identification and quantification of PvbXa were performed using XPS analysis. In particular, the existence of sulphide was observed at around 20 wt.%. This was calculated from the XPS survey scans of PvbXa (Figure 3). Figure 3 shows the binding energies of the atom in XPS survey scans of PvbXa.

### 2.6. VA-dSPµE Procedure

The experimental steps of the VA-dSPµE procedure for the efficient and selective extraction of Cd(II) are as follows: 10 mL of the digested samples or 200 mL of water samples were transferred into conical tubes. Then, 125 mg of synthesized PvbXa was added to the tubes. In the third step, pH was adjusted to 4.5 by the acetate buffer to facilitate the adsorption of Cd(II) ions in the solution onto the PvbXa. Then the tubes were vortexed at 200 rpm for 8 min to confirm the effective distribution of the PvbXa added to the sample solution, and the tubes were then centrifuged for 5 min at 4000 rpm to collect the snow sorbent at the bottom of the tube. The aqueous portion was drained by decantation. To obtain the Cd(II) ions adsorbed onto the solid back in the measurement solution, 1250 µL of EtOH was added and vortexed for 120 s. Finally, Cd determination was performed by injecting the death solution into the atomization section of the FAAS. All experimental steps were carried out together with the reagent blank.

### 2.7. Factorial Design

Factorial designs at three levels are mostly utilized by screening the impact of variables on the response and removing those variables that are not significant [34]. The present study coded low (−1), middle (0), and high (+1) levels, which were optimized using a central composite design (CCD) (Table 1). The 30 experimental runs drawn in the factorial design used four variables, namely pH, sorbent amount in mg, mixing time (min), and sample volume (see Table 2).

## 3. Results

### 3.1. Optimized Factorial Design

#### 3.1.1. Response Surface Plots

Three-dimensional 3D plots were applied for the cadmium. These 3D plots are suitable for recognizing the interaction among the factors and measuring the optimum conditions of each factor. Surface plots determine the influence of two factors [35]. Figure 4a shows the response surface drawn for pH and the sorbent amount, and the plot depicts that the recovery of Cd was high when increasing the pH from 2 to 4 and the sorbent amount from 20 to 120 mg, while a reduction was observed with the continuous increase in the level of pH and sorbent amount from 4 to 10 and 120 to 200 mg, respectively. Therefore, the maximum recovery of the analyte was achieved at pH 4 and a sorbent amount of 120 mg. Figure 4b shows the response surface established for pH and mixing time, and the plot depicts that the recovery of the analyte increased with an increased mixing time from 2 to 8 min and then decreased with a continuous increase in the mixing time from 8 to 20 min at the optimized pH level of 4. Figure 5a shows the response surface obtained for the pH and sample volume, and the plot describes that the recovery of the analyte decreased with an increasing sample volume from 25 to 250 mL at the optimized pH level of 4. Figure 5b shows the response surface established for the sorbent amount and mixing time, and the plot represents that the recovery of the analyte increased with an increasing sorbent amount and mixing time from 20 to 120 mg and 2 to 8 min, respectively, then decreased with a continuous increase in the level of sorbent amount and mixing time. Figure 6a shows the response obtained for the sorbent amount and sample volume, and the plot represents that the recovery of the analyte decreases with an increasing sample volume at the optimized sorbent amount of 120 mg. Figure 6b shows the response surface obtained for mixing time and sample volume, and the plot represents that the recovery of the analyte decreased with an increasing sample volume at the optimized low mixing time of 8 min.

#### 3.1.2. Analysis of Variance ANOVA

After conducting the CCD model, ANOVA was utilized to evaluate the significance level of the proposed model (see Appendix A). The model is significant and fit if the *p*-value is <0.05 [35]. The *p*-value was 0.0 to 1.0 and therefore variables B, C, D AA, BB, CC, and DD were significant and the model was fit as the *p*-value was <0.05. Other variables were not significant as *p*-values were greater than 0.05.

#### 3.1.3. Pareto Chart

The effects of variables were estimated utilizing the Pareto chart of standardized effects [36]. The plot is linked to the absolute value of the probable standardized influence and vertical line (2.160) with statistically significant effects at the 95% confidence level (see Figure 7). According to the obtained results, the sorbent amount (B), mixing time (C), sample volume (D), and combined factors of DD, AA, CC, and BB were fit in the model, and these variables showed significant effects on the recovery of cadmium. Other variables did not fit and presented no significant effects on the result of Cd in the samples.

#### 3.1.4. Normal and Half Normal Plots

These plots are useful to explain the factors and indicate significant and non-significant levels of the variables [37]. According to the obtained results, the sorbent amount (B), mixing time (C), sample volume (D), and combined factors of CC, AA, DD, and BB fit in the model (see Figure 8a,b), and these variables showed significant effects on the recovery of cadmium, while other variables were not significant due to the lack of model fit.

#### 3.1.5. Residual Plots

The residual plots are useful to measure the excellence of the model, and residuals will normally be scattered if the model is fit and significant [38]. Figure 9a describes the normal probability versus internally standardized residuals, and there is no significant deviance from the straight line. The histogram in Figure 9b shows the vertical bar line distribution of negative and positive values, and this plot indicates that the model is normally distributed for both significant and insignificant values. Furthermore, the residuals plot versus the fitted values (shown in Figure 9c,d) demonstrate that residuals were dispersed with random behavior on both sides of the critical line, which shows good agreement between the results achieved by the model and indicates that the model is statistically significant.

### 3.2. Effects of Parameter for Recovery of Cadmium

Different working parameters were used, and their impact on the extraction recovery of Cd was determined. The impacts of pH on the extraction recovery of Cd in food and water samples were studied with a range of 2 to 10 (see Appendix A). The recovery of Cd was attained under acidic pH environments at pH 4. Further study was performed by keeping the value of pH of the sample at 4 for the recovery of cadmium.

The amount of sorbent plays an important role in the extraction of cadmium, and the sorbent amount was studied in a range of 25 to 200 mg (see Appendix A). It was observed that the recovery of Cd increased when the amount of sorbent was increased to 125 mg; after that, recovery decreased, which may be due to the active site being achieved at 125 mg so a greater sorbent amount could improve the recovery of Cd. Therefore, 125 mg of sorbent was the amount optimized for further study.

There are many types of mixing procedures reported; therefore, in the present study, different types of mixing methods were applied to mix the sorbent, solvent, and sample. These methods are vortex, sonication, hand mixing, and orbital shaking and were applied in this study to assess their efficiency in the recovery of Cd (see Appendix A). The results indicated that vortex mixing produced better extraction recovery of cadmium compared to other mixing procedures, and vortex was therefore optimized for further work. Different mixing times were studied, with a range of 2.5 to 20 min (see Appendix A), and a mixing time of 7.5 min showed better recovery of cadmium. The results revealed that a higher mixing time did not improve the recovery of Cd. Therefore, a mixing time of 7.5 min was selected for further work.

For better extraction recovery of cadmium, the selection of the eluent solvent is important, and in the present study, five different types of solvents were used, namely acetonitrile (ACN), acetone, heptanol, MeOH, and EtOH (see Appendix A). The results indicated that EtOH exhibited better recovery of cadmium compared to the other solvents and was therefore adjusted for further work. The volume of the EtOH solvent was studied in the range of 200 to 1400 µL and the results showed that the extraction recovery of cadmium increased with an increased volume of the EtOH solvent of up to 1250 µL (see Appendix A). Therefore, an EtOH volume of 1250 µL was adjusted for the study. The eluent time studied ranged from 20 to 180 s, and the results showed that the extraction recovery of cadmium increased with an increased eluent time of up to 120 s (see Appendix A). Therefore, an eluent time of 120 s was adjusted for the study.

Different numbers of resumes of the sorbent were studied and ranged from 1 to 20; it was observed that the recovery of Cd slightly decreased when the number of resumes of the solvent increased, therefore a low resume of the sorbent was favorable for better recovery of cadmium (see Appendix A) and was chosen for further research work.

The volume of the sample shows a major effect on the interactions between the extracting and analyte ions. The volume of the sample varied from 25 to 250 mL (see Appendix A). The results showed that when the volume of the sample was low, the maximum extraction recovery of Cd was obtained, but when the volume of the sample increased, the recovery of Cd decreased. Therefore, maximum recovery was obtained at a sample volume of 200 mL to obtain a high preconcentration factor.

### 3.3. Matrix Effect

The tolerance limit of concomitant ions is definitive as ion concentrations cause a relative error of analyte ions lower than 5% (Appendix A). It was determined that concomitant ions did not disturb the recovery of analyte ions. The results show that VA-dSPµE of cadmium from samples via FAAS can be easily applied to highly saline samples. The present VA-dSPµE method is highly selective for uses in water, environmental samples, and food samples for the extraction of Cd ions.

### 3.4. Analytical Figure of Merits

The analytical performance of the VA-dSPµE method was assessed using LOD, LOQ, LR, extraction recovery (ER), intraday and intraday precision, PF, and EF. Blank solutions including eleven different samples were used for the calculation of parameters using the VA-dSPµE method. LOD was found to be 0.06 µg L^−1^ and LOQ was 0.20 µg L^−1^. The calibration curve was linear in 0.20–150 µg L^−1^ and R^2^ was 0.995. Intraday precision (N = 5) was found to be 2.4, 3.1, and 3.6% for 1, 50, and 100 µg L^−1^ of Cd(II), respectively. Inter-day precision (N = 5) was found to be 2.7, 3.7, and 4.3% for 1, 50, and 100 µg L^−1^ of Cd(II). The PF and EF of the VA-dSPµE method were 160 and 100, respectively (see Table 3). The EF was calculated by using the ratio of the direct calibration curves’ slopes obtained with and without the VA-dSPµE method. The PF was calculated from the ratio of the initial volume to the final volume.

### 3.5. Validation and Applications of Present Method

Validation of the present VA-dSPµE method was tested with three different certified reference materials (INCT-TL-1 Tea leaves, SRM-1547 Peach leaves, SRM-1643e Trace element in water) (Table 4). The results were definitive with certified values, no differences were obtained, and 95–98% recovery was found.

The present VA-dSPµE method was utilized in various water samples comprising tap water, well water, bottled water, and cold spring water. To measure the accuracy of the present VA-dSPµE method, the samples were spiked with standard solutions. Regarding 75 ng mL^−1^ of spiked Cd(II), recovery of 92–98% was obtained and RSD values were found to be below 3.0% (Table 5). The present VA-dSPµE method was utilized for microwave-digested food comprising apple, spinach, salad, tomatoes, onion, oat, corn, aubergine, wheat, rice, and mushroom samples. Cadmium levels in the measured samples were found in µg g^−1^ levels with a 1.4–4.3% RSD value (Table 6), and the achieved result was found to be within the WHO limits [14]. There are not any risks in the consumption of the analyzed food samples with respect to human health. However, food samples should be analyzed more often with respect to cadmium.

## 4. Discussion

In the present study, a new polyvinyl benzyl xanthate (PvbXa) adsorbent-based VA-dSPµE procedure was developed for the determination of Cd from food and water samples via FAAS. The value of pH is a critical factor that may significantly affect the efficiency of extraction and metal separation through the microextraction procedure comprising the prior formation of the complex, which has sufficient hydrophobicity [39]. The metal–ligand interaction for complex formation and efficiency of extraction is directly dependent on the pH level of the solution. A factorial design was utilized for screening the impact of the variables on the response and removing the variables that are not significant [34]. The CCD model is a powerful tool for analyzing the optimum level of parameters used in the analysis of Cd in food and water samples. Different plots were obtained in the factorial design such as 3D plots, Pareto charts, residual plots, and *p*-values and F-values in ANOVA. These plots are very effective in determining the optimum levels of parameters, and their significance levels were described by the CCD model. Many methods reported for use in the determination of Cd in the present work were compared with other described procedures. Sorouraddin et al. developed the reverse-phase dispersive LLME procedure for the extraction of Cd ions in certain cosmetic products such as different lipsticks and cream samples, and recovery of 88 to 98% was achieved [26]. Shishov et al. analyzed Cd in vegetable oil by reverse-phase DLLME using DES as a solvent with the help of voltammetry and 85% recovery of Cd was reported. The effect of the DES solvent, extraction time, centrifugation time, and interference study were also studied [27]. Xue et al. determined Cd in tea, water, distilled spirits, and juice by successive homogeneous liquid-liquid microextraction, and 77.0–92.3% recovery was observed [40]. Sun et al. compared the recovery of Cd(II) by FAAS using UA-DLLME and the results indicated that 96.7 to 113.6% was achieved in glutinous rice, polished rice, and brown rice samples [41]. Shamsipur et al. prepared the Natural DES-based ultrasound-vortex-assisted DLLME determination of trace levels of Cd ions in food and water. Linearity was found in the range of 0.001–7.5 µgL^−1^, (R^2^ = 0.995). The planned method provided a good LOD of 0.37 × 10^−4^ and LOQ of 1.24 × 10^−4^ µgL^−1^. The PF factor was obtained at 125 and RSD% was 2.65%. Moreover, 95–99% recovery of Cd was reported by the proposed procedure [42]. Elik and Altunay reported the MIL-DLLME procedure for the recovery of Cd ions from different water and food samples. The dynamic range for recovery of Cd(II) was 2–700 ng mL^−1^. The LOD was 0.6 and the LOQ was 2.0 ng mL^−1^. RSD% was found to be 1.5%, while EF was 172, and 98% recovery of Cd was reported [43]. Yang et al. developed graphene oxide from pencils for solid-phase microextraction of Cd by using GF-AAS. The calibration curve for Cd ions was linear in the range of 0.04–0.26 μg L^−1^ with a LOD of 0.005 μg L^−1^. The RSD was 2.1% with an EF value of 25. The recovery of Cd in tap water, river water, and pond water ranged from 94 to 105% [44]. There are many other methods reported for the determination of Cd in different samples using different extraction methods [45,46,47,48,49,50,51,52] (Table 7). The analytical efficacy of the present optimized procedure was compared with many previously reported techniques for the analysis of Cd and is described in Table 5 and Table 6. Among the reported methods, the present method displayed a lower RSD% and extraction time. Specifically, a good linearity range, LOD, LOQ, and recovery were obtained as compared to other ETAAS and HR-CS-FAAS methods, which are more sensitive than FAAS. In our results, the linearity range, EF, LOD, RSD%, and LOQ values of the optimized procedure for the extraction of Cd were better than other reported procedures. Toxic reagents are not required in the present procedure, which provides an important advantage over other procedures.

## 5. Conclusions

A simple and highly sensitive method was developed on the basis of VA-dSPµE combined with FAAS and utilized for the extraction of Cd in water and food samples with the use of a sulphur-rich sorbent, PvbXa. The contribution of eluent time, pH, sorbent level, mixing type and time, resume number, and sample volume were adjusted by the CCD model used. The proposed method is effectively utilized to assess Cd and showed good extraction recovery from food and water samples. Comparison of the VA-dSPµE method with the reported values is specified in Table 7. This method has certain advantages such as low LOD and RSD, short time of extraction, good linear range, and better preconcentration factor according to the reported values. The tolerance limit of concomitant ions was found at a higher level. Thus, the present VA-dSPµE method may be easily utilized with highly saline samples and complex matrice media for the extraction and examination of Cd.

## Figures and Tables

**Figure 1 foods-12-03620-f001:**
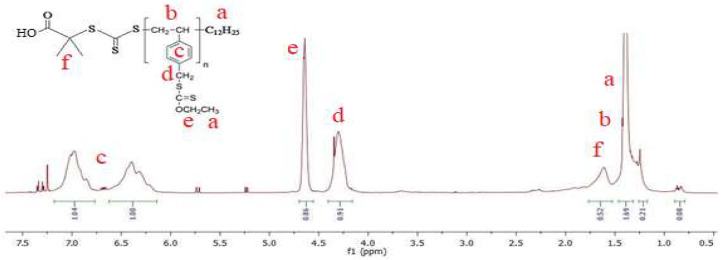
^1^H NMR spectrum of PvbXa with signals marked with functional groups.

**Figure 2 foods-12-03620-f002:**
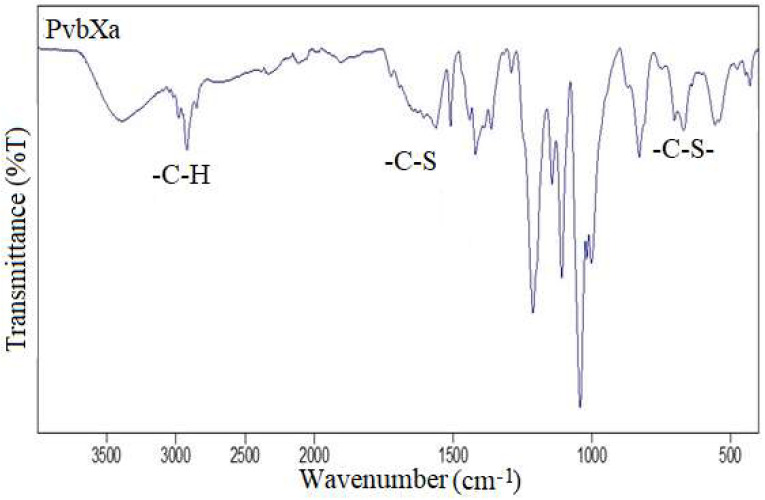
FTIR spectrum of PvbXa.

**Figure 3 foods-12-03620-f003:**
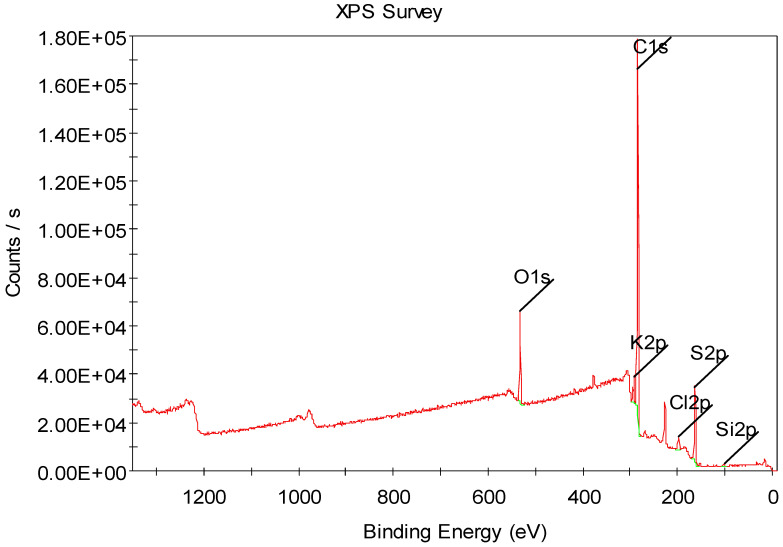
XPS survey scans of PvbXa.

**Figure 4 foods-12-03620-f004:**
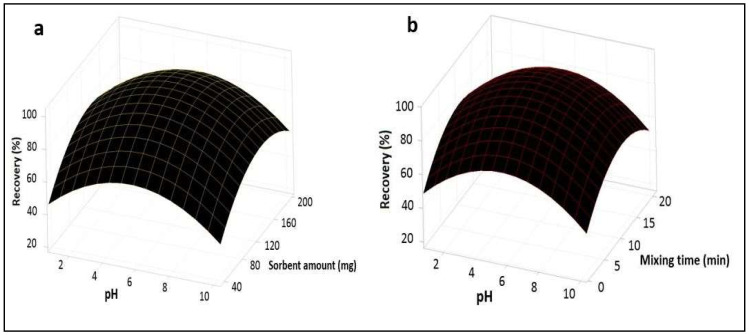
Response surface plots (**a**,**b**).

**Figure 5 foods-12-03620-f005:**
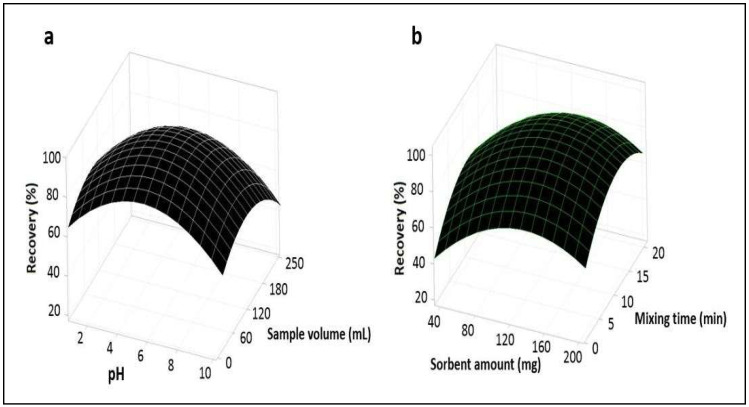
Response surface plots (**a**,**b**).

**Figure 6 foods-12-03620-f006:**
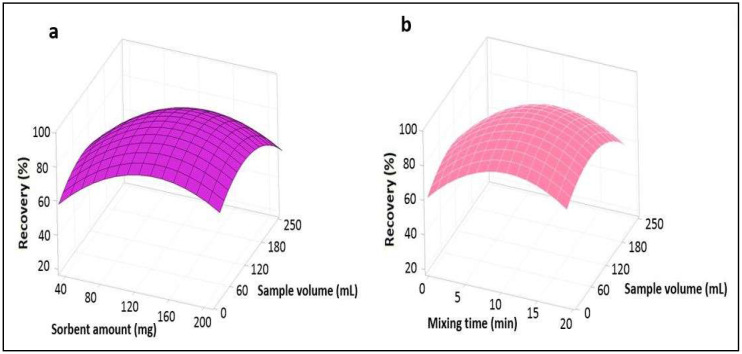
Response surface plots (**a**,**b**).

**Figure 7 foods-12-03620-f007:**
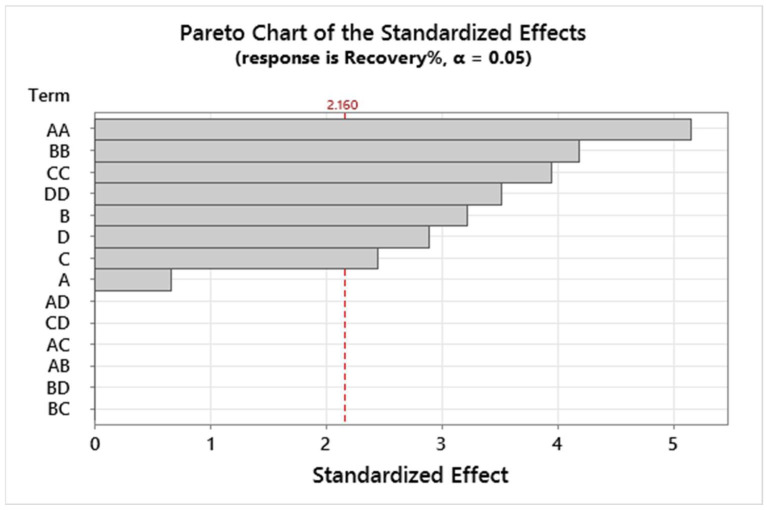
Pareto chart of the variables.

**Figure 8 foods-12-03620-f008:**
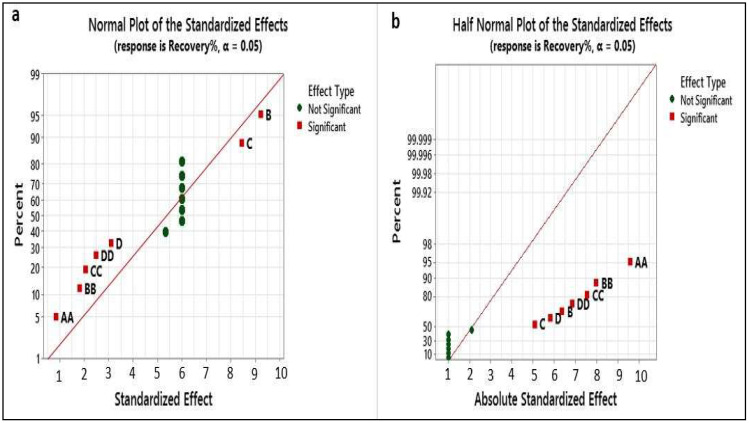
(**a**) Normal probability plot and (**b**) half normal plot.

**Figure 9 foods-12-03620-f009:**
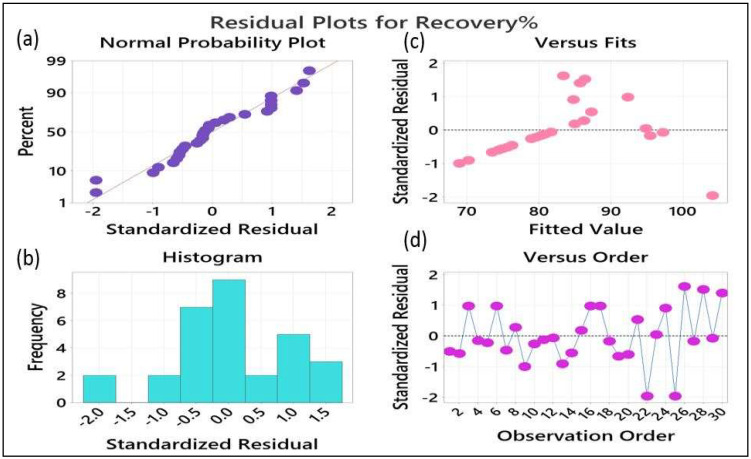
Residuals plots (**a**–**d**).

**Table 1 foods-12-03620-t001:** Levels of factors used in experimental design.

Variables	Symbol	Low (−1)	Middle (0)	High (+1)
pH	A	2	4	10
Sorbent amount (mg)	B	25	125	200
mixing time (min)	C	2	8	20
Sample volume (mL)	D	25	150	250

**Table 2 foods-12-03620-t002:** Experiments design for factorial design (central composite design).

Run	A	B	C	D	Recovery %
1	−1	1	−1	−1	71.16
2	−1	−1	1	1	72.46
3	0	1	0	0	76.48
4	1	1	−1	−1	79.93
5	1	−1	1	1	80.10
6	0	−1	0	0	86.54
7	−1	1	−1	1	84.56
8	−1	1	1	−1	87.00
9	1	−1	−1	1	72.45
10	1	−1	1	−1	81.64
11	−1	1	1	1	80.58
12	−1	1	−1	−1	83.32
13	−1	−1	−1	1	87.77
14	1	1	−1	1	83.19
15	1	1	1	−1	85.50
16	0	0	0	0	96.87
17	0	0	0	0	96.87
18	−1	−1	1	−1	79.75
19	1	−1	1	1	71.42
20	1	−1	−1	−1	72.83
21	−1	0	0	0	78.65
22	0	0	1	0	86.93
23	0	0	1	0	95.10
24	1	0	0	0	85.76
25	0	0	0	0	96.75
26	0	−1	0	0	87.63
27	0	1	0	0	87.30
28	0	0	0	1	87.55
29	−1	0	0	0	88.75
30	0	0	−1	0	87.25

**Table 3 foods-12-03620-t003:** Analytical performance characteristics of the VA-dSPµE-FAAS for determination of Cd.

Parameters	Values
Regression equation	A = 0.092 [Cd, µg L^−1^] + 0.0023
R^2^	0.995
Linear range, µg L^−1^	0.2–150
LOD, µg L^−1^	0.06
LOQ, µg L^−1^	0.20
Intra-day precision for 1, 50 and 100 µg L^−1^ of Cd(II) (N = 5)	2.4, 3.1 and 3.6
Inter-day precision for 1, 50 and 100 µg L^−1^ of Cd(II) (N = 3 × 5)	2.7, 3.7 and 4.3
Average Recovery (%)	97.1
PF	160

**Table 4 foods-12-03620-t004:** Results of the validation study of the VA-dSPµE-FAAS method (N = 5).

Reference Material	Certified Value ^a,b^	Experimental Value ^a,b^	Recovery (%)	t-exp
INCT-TL-1 Tea leaves	30.0 ± 4.0 ^a^	29.4 ± 1.6	98 ± 1	0.86
SRM-1547	2.61 ± 2.2 ^a^	2.50 ± 0.57	96 ± 3	0.43
SRM-1643e	6.568 ± 0.073 ^b^	6.239 ± 0.651	95 ± 2	1.13

^a^: µg kg^−1^, ^b^: µg L^−1^.

**Table 5 foods-12-03620-t005:** Application results of the VA-dSPµE-FAAS method to water samples (N = 3).

Samples	Spiked (µg L^−1^)	Obtained Value (µg L^−1^)	Recovery (%)
Tap water	-	4.2 ± 0.2 ^a^	-
	75	76.2 ± 2.9	96 ± 4
Hot spring water	-	n.d *	-
	75	73.5 ± 4.3	98 ± 1
Well water	-	8.9 ± 1.1	-
	75	82.4 ± 5.7	98 ± 3
Bottled water	-	n.d	-
	75	70.5 ± 2.9	94 ± 5
Cold spring water	-	n.d	-
	75	69 ± 2.5	92 ± 6

* could not be determined. ^a^ Mean ± standard deviation. n.d = could not be determinded.

**Table 6 foods-12-03620-t006:** Application results of the VA-dSPµE-FAAS method to food samples (N = 3).

Samples	Obtained Value (µg g^−1^)	RSD (%)
Apple	0.47 ± 0.02 ^a^	4.3
Spinach	0.89 ± 0.02	2.2
Salad	1.15 ± 0.03	2.6
Tomatoes	0.33 ± 0.01	3.0
Onion	2.08 ± 0.06	2.8
Oat	0.71 ± 0.01	1.4
Corn	0.53 ± 0.02	3.7
Aubergine	0.18 ± 0.01	3.5
Wheat	0.95 ± 0.03	3.2
Mushroom	1.14 ± 0.04	3.5
Black rice	0.80 ± 0.05	3.1
Brown rice	0.65 ± 0.02	3.7
White rice	0.95 ± 0.06	2.8

^a^ Mean ± standard deviation.

**Table 7 foods-12-03620-t007:** Comparison of the present method with reported methods for determination of cadmium.

Name of Instruments	Method	LOD µg L^−1^	LOQ µg L^−1^	LR µg L^−1^	ER (Time min)	PF, EF	RSD%	References
FAAS	RP-DLLME	0.006		0.02–50	5.0	39	6.0	[27]
FAAS	RP-DLLME	0.3	1.0	1.0–175	10	20.8	6.5	[28]
FAAS	DMSPE	0.96		5–400	115	5.0	1.54	[35]
GFAAS	SHLLME	6.0	20.0	20–600	2.0	41.5	2.9	[40]
FAAS	UADLLME	0.69	2.08	0.1–55	5.0	31	2.30	[41]
FAAS	SPE	0.216	0.648	100–1000	10	40	4.01	[45]
HR-CS -FAAS	EA-SS-LPME	6.8	22.8	20–1500	8.0	2.6	1.69	[46]
FAAS	UDDLLμE	0.046	0.300	1–50	15.0	78.2	2.0	[47]
FAAS	UA-DLLME-DES	0.08	0.25	0.50–8.0	1.0		2.9	[48]
FAAS	USAEME	0.39	1.33	2.5–50	4.0	21	3.9	[49]
FAAS	HDES-LPME	1.6	5.0	50- 100	5.0	43	3.3	[50]
SQT-FAAS	DESMNF-LPME	0.25	0.84	1.0–30	2.0	73.3	5.0	[51]
ETAAS	MIL-DLLME	0.084	0.28	0.28–4.75	1.0	220	6.0	[52]
FAAS	VA-dSPµE	0.06	0.20	0.20–150	5.0	160	4.3	Present work

Successive homogeneous liquid-liquid microextraction (SHLLME), Ultrasound-assisted dispersive liquid–liquid microextraction (UADLLME), Reversed-phase dispersive liquid–liquid microextraction, Vortex-assisted- Ultrasound-Dispersive Liquid–Liquid Microextraction (VE-UA-DLLME), Magnetic ionic based dispersive liquid–liquid microextraction (MIL-DLLME), Effervescent tablet-assisted switchable solvent based liquid phase microextraction (EA-SS-LPME), High-resolution continuum source flame atomic absorption spectrometry (HR-CS-FAAS), Solid-phase microextraction (SPME), Ultrasonic-assisted dual dispersive liquid–liquid microextraction (UDDLLμE), Dispersive micro solid-phase extraction (DMSPE), Deep eutectic solvent (DES), Ultrasound-assisted emulsification Microextraction (USAEME), Hydrophobic deep eutectic solvent based liquid phase microextraction (HDES-LPME), Slotted quartz tubeflame atomic absorption spectrophotometry (SQT-FAAS), deep eutectic solvent-based magnetic nanofluid liquid phase microextraction method (DESMNF-LPME), Electro-thermal atomic absorption spectrophotometer (ETAAS), Magnetic ionic liquids based dispersive liquid–liquid microextraction (MIL-DLLME), Limit of detection (LOD), Limit of quantification (LOQ), Linear range (LR), Extraction recovery time (ER), Pre-concentration factor (PF), Enrichment factor (EF), Relative standard deviation (RSD).

## Data Availability

The data used to support the findings of this study can be made available by the corresponding author upon request.

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
