# Peer review of "Ultra-Sensitive Determination of Cadmium in Food and Water by Flame-AAS after a New Polyvinyl Benzyl Xanthate as an Adsorbent Based Vortex Assisted Dispersive Solid-Phase Microextraction: Multivariate Optimization"

_foods, 2023, doi:10.3390/foods12193620_

Round 1
Reviewer 1 Report
The paper entitled ‘Ultra-sensitive determination of cadmium in food and water by flame-AAS after a new polyvinyl benzyl xanthate as an adsorbent based vortex assisted dispersive solid-phase microextraction: multivariate optimization’ reported a new polyvinyl benzyl xanthate (PvbXa) as adsorbent based vortex assisted dispersive solid-phase microextraction (VA-dSPµE) procedure was developed for the determination of cadmium (Cd) from food and water samples via flame atomic absorption spectrophotometry. This work in interesting. I have no specific concern but it is mandatory to add the manuscript in these point:
1. In Figure 2, the interval between the horizontal coordinates of the chart are uneven. All the bonds in FTIR should be assigned.
2. In Table 2, experiments design for factorial design, the Run 29 with a recovery at 97.0 %, however, subsequent research lacks corresponding validation.
Author Response
Formun Üstü
Author's Reply to the Review Report (Reviewer 1)
Please provide a point-by-point response to the reviewer’s comments and either enter it in the box below or upload it as a Word/PDF file. Please write down "Please see the attachment." in the box if you only upload an attachment. An example can be found here.
* Author's Notes to Reviewer
FileEditViewInsertFormatToolsTableHelp
Paragraph
P
0 WORDS
Word / PDF
or
Formun Altı
Formun Üstü
Review Report Form
Open Review
(x) I would not like to sign my review report
( ) I would like to sign my review report
Quality of English Language
( ) I am not qualified to assess the quality of English in this paper
( ) English very difficult to understand/incomprehensible
( ) Extensive editing of English language required
( ) Moderate editing of English language required
( ) Minor editing of English language required
(x) English language fine. No issues detected
|
Yes |
Can be improved |
Must be improved |
Not applicable |
|
|
Does the introduction provide sufficient background and include all relevant references? |
(x) |
( ) |
( ) |
( ) |
|
Are all the cited references relevant to the research? |
( ) |
(x) |
( ) |
( ) |
|
Is the research design appropriate? |
(x) |
( ) |
( ) |
( ) |
|
Are the methods adequately described? |
(x) |
( ) |
( ) |
( ) |
|
Are the results clearly presented? |
(x) |
( ) |
( ) |
( ) |
|
Are the conclusions supported by the results? |
(x) |
( ) |
( ) |
( ) |
Comments and Suggestions for Authors
The paper entitled ‘Ultra-sensitive determination of cadmium in food and water by flame-AAS after a new polyvinyl benzyl xanthate as an adsorbent based vortex assisted dispersive solid-phase microextraction: multivariate optimization’ reported a new polyvinyl benzyl xanthate (PvbXa) as adsorbent based vortex assisted dispersive solid-phase microextraction (VA-dSPµE) procedure was developed for the determination of cadmium (Cd) from food and water samples via flame atomic absorption spectrophotometry. This work in interesting. I have no specific concern but it is mandatory to add the manuscript in these point:
- In Figure 2, the interval between the horizontal coordinates of the chart are uneven. All the bonds in FTIR should be assigned.
The FTIR scales are accustomed standard scales, there comes from “Bruker FTIR instrument”. The characteristic bands were assigned as shown below:
- In Table 2, experiments design for factorial design, the Run 29 with a recovery at 97.0 %, however, subsequent research lacks corresponding validation.
Response: Thank you for valuable comments. We have redrawn the Table 2 and also checked their corresponding validation, hope now it will be acceptable.
Formun Üstü
Journal
Foods (ISSN 2304-8158)
Manuscript ID
foods-2578312
Type
Article
Title
Ultra-sensitive determination of cadmium in food and water by flame-AAS after a new polyvinyl benzyl xanthate as an adsorbent based vortex assisted dispersive solid-phase microextraction: multivariate optimization
Authors
Nail Altunay , Baki Hazer * , Muhammad Farooque Lanjwani , Mustafa Tuzen
Section
Food Analytical Methods
Abstract
A new polyvinyl benzyl xanthate (PvbXa) as adsorbent based vortex assisted dispersive solid-phase microextraction (VA-dSPµE) procedure was developed for the determination of cadmium (Cd) from food and water samples via flame atomic absorption spectrophotometry (FAAS). Synthesized PvbXa was characterized with 1H Nuclear magnetic resonance (NMR) Spectroscopy, Fourier Transform Infrared Spectroscopy (FTIR) and X-Ray Photoelectron Spectroscopy (XPS). The different parameters pH, sample volume, mixing type and time, sorbent amount, eluent time were optimized using standard analytical method. The optimized method for assessment of Cd in food and water samples shows good reliability. The good optimum conditions were found like 0.20–150 µg L-1 linear range, 0.06 µg L-1 LOD, 0.20 µg L-1 LOQ, 4.3 RSD %, and 200 preconcentration factor. The statistically experimental for variables were utilized using central composite design (CCD).
Formun Altı
Formun Üstü
Formun Üstü
Author's Reply to the Review Report (Reviewer 1)
Please provide a point-by-point response to the reviewer’s comments and either enter it in the box below or upload it as a Word/PDF file. Please write down "Please see the attachment." in the box if you only upload an attachment. An example can be found here.
* Author's Notes to Reviewer
FileEditViewInsertFormatToolsTableHelp
Paragraph
P
0 WORDS
Word / PDF
or
Formun Altı
Formun Üstü
Review Report Form
Open Review
(x) I would not like to sign my review report
( ) I would like to sign my review report
Quality of English Language
( ) I am not qualified to assess the quality of English in this paper
( ) English very difficult to understand/incomprehensible
( ) Extensive editing of English language required
( ) Moderate editing of English language required
( ) Minor editing of English language required
(x) English language fine. No issues detected
|
Yes |
Can be improved |
Must be improved |
Not applicable |
|
|
Does the introduction provide sufficient background and include all relevant references? |
(x) |
( ) |
( ) |
( ) |
|
Are all the cited references relevant to the research? |
( ) |
(x) |
( ) |
( ) |
|
Is the research design appropriate? |
(x) |
( ) |
( ) |
( ) |
|
Are the methods adequately described? |
(x) |
( ) |
( ) |
( ) |
|
Are the results clearly presented? |
(x) |
( ) |
( ) |
( ) |
|
Are the conclusions supported by the results? |
(x) |
( ) |
( ) |
( ) |
Comments and Suggestions for Authors
The paper entitled ‘Ultra-sensitive determination of cadmium in food and water by flame-AAS after a new polyvinyl benzyl xanthate as an adsorbent based vortex assisted dispersive solid-phase microextraction: multivariate optimization’ reported a new polyvinyl benzyl xanthate (PvbXa) as adsorbent based vortex assisted dispersive solid-phase microextraction (VA-dSPµE) procedure was developed for the determination of cadmium (Cd) from food and water samples via flame atomic absorption spectrophotometry. This work in interesting. I have no specific concern but it is mandatory to add the manuscript in these point:
- In Figure 2, the interval between the horizontal coordinates of the chart are uneven. All the bonds in FTIR should be assigned.
The FTIR scales are accustomed standard scales, there comes from “Bruker FTIR instrument”. The characteristic bands were assigned as shown below:
- In Table 2, experiments design for factorial design, the Run 29 with a recovery at 97.0 %, however, subsequent research lacks corresponding validation.
Response: Thank you for valuable comments. We have redrawn the Table 2 and also checked their corresponding validation, hope now it will be acceptable.
Formun Üstü
Journal
Foods (ISSN 2304-8158)
Manuscript ID
foods-2578312
Type
Article
Title
Ultra-sensitive determination of cadmium in food and water by flame-AAS after a new polyvinyl benzyl xanthate as an adsorbent based vortex assisted dispersive solid-phase microextraction: multivariate optimization
Authors
Nail Altunay , Baki Hazer * , Muhammad Farooque Lanjwani , Mustafa Tuzen
Section
Food Analytical Methods
Abstract
A new polyvinyl benzyl xanthate (PvbXa) as adsorbent based vortex assisted dispersive solid-phase microextraction (VA-dSPµE) procedure was developed for the determination of cadmium (Cd) from food and water samples via flame atomic absorption spectrophotometry (FAAS). Synthesized PvbXa was characterized with 1H Nuclear magnetic resonance (NMR) Spectroscopy, Fourier Transform Infrared Spectroscopy (FTIR) and X-Ray Photoelectron Spectroscopy (XPS). The different parameters pH, sample volume, mixing type and time, sorbent amount, eluent time were optimized using standard analytical method. The optimized method for assessment of Cd in food and water samples shows good reliability. The good optimum conditions were found like 0.20–150 µg L-1 linear range, 0.06 µg L-1 LOD, 0.20 µg L-1 LOQ, 4.3 RSD %, and 200 preconcentration factor. The statistically experimental for variables were utilized using central composite design (CCD).
Formun Altı
Formun Üstü
Formun Üstü
Author's Reply to the Review Report (Reviewer 1)
Please provide a point-by-point response to the reviewer’s comments and either enter it in the box below or upload it as a Word/PDF file. Please write down "Please see the attachment." in the box if you only upload an attachment. An example can be found here.
* Author's Notes to Reviewer
FileEditViewInsertFormatToolsTableHelp
Paragraph
P
0 WORDS
Word / PDF
or
Formun Altı
Formun Üstü
Review Report Form
Open Review
(x) I would not like to sign my review report
( ) I would like to sign my review report
Quality of English Language
( ) I am not qualified to assess the quality of English in this paper
( ) English very difficult to understand/incomprehensible
( ) Extensive editing of English language required
( ) Moderate editing of English language required
( ) Minor editing of English language required
(x) English language fine. No issues detected
|
Yes |
Can be improved |
Must be improved |
Not applicable |
|
|
Does the introduction provide sufficient background and include all relevant references? |
(x) |
( ) |
( ) |
( ) |
|
Are all the cited references relevant to the research? |
( ) |
(x) |
( ) |
( ) |
|
Is the research design appropriate? |
(x) |
( ) |
( ) |
( ) |
|
Are the methods adequately described? |
(x) |
( ) |
( ) |
( ) |
|
Are the results clearly presented? |
(x) |
( ) |
( ) |
( ) |
|
Are the conclusions supported by the results? |
(x) |
( ) |
( ) |
( ) |
Comments and Suggestions for Authors
The paper entitled ‘Ultra-sensitive determination of cadmium in food and water by flame-AAS after a new polyvinyl benzyl xanthate as an adsorbent based vortex assisted dispersive solid-phase microextraction: multivariate optimization’ reported a new polyvinyl benzyl xanthate (PvbXa) as adsorbent based vortex assisted dispersive solid-phase microextraction (VA-dSPµE) procedure was developed for the determination of cadmium (Cd) from food and water samples via flame atomic absorption spectrophotometry. This work in interesting. I have no specific concern but it is mandatory to add the manuscript in these point:
- In Figure 2, the interval between the horizontal coordinates of the chart are uneven. All the bonds in FTIR should be assigned.
The FTIR scales are accustomed standard scales, there comes from “Bruker FTIR instrument”. The characteristic bands were assigned as shown below:
- In Table 2, experiments design for factorial design, the Run 29 with a recovery at 97.0 %, however, subsequent research lacks corresponding validation.
Response: Thank you for valuable comments. We have redrawn the Table 2 and also checked their corresponding validation, hope now it will be acceptable.
Formun Üstü
Journal
Foods (ISSN 2304-8158)
Manuscript ID
foods-2578312
Type
Article
Title
Ultra-sensitive determination of cadmium in food and water by flame-AAS after a new polyvinyl benzyl xanthate as an adsorbent based vortex assisted dispersive solid-phase microextraction: multivariate optimization
Authors
Nail Altunay , Baki Hazer * , Muhammad Farooque Lanjwani , Mustafa Tuzen
Section
Food Analytical Methods
Abstract
A new polyvinyl benzyl xanthate (PvbXa) as adsorbent based vortex assisted dispersive solid-phase microextraction (VA-dSPµE) procedure was developed for the determination of cadmium (Cd) from food and water samples via flame atomic absorption spectrophotometry (FAAS). Synthesized PvbXa was characterized with 1H Nuclear magnetic resonance (NMR) Spectroscopy, Fourier Transform Infrared Spectroscopy (FTIR) and X-Ray Photoelectron Spectroscopy (XPS). The different parameters pH, sample volume, mixing type and time, sorbent amount, eluent time were optimized using standard analytical method. The optimized method for assessment of Cd in food and water samples shows good reliability. The good optimum conditions were found like 0.20–150 µg L-1 linear range, 0.06 µg L-1 LOD, 0.20 µg L-1 LOQ, 4.3 RSD %, and 200 preconcentration factor. The statistically experimental for variables were utilized using central composite design (CCD).
Formun Altı
Formun Üstü
Formun Üstü
Author's Reply to the Review Report (Reviewer 1)
Please provide a point-by-point response to the reviewer’s comments and either enter it in the box below or upload it as a Word/PDF file. Please write down "Please see the attachment." in the box if you only upload an attachment. An example can be found here.
* Author's Notes to Reviewer
FileEditViewInsertFormatToolsTableHelp
Paragraph
P
0 WORDS
Word / PDF
or
Formun Altı
Formun Üstü
Review Report Form
Open Review
(x) I would not like to sign my review report
( ) I would like to sign my review report
Quality of English Language
( ) I am not qualified to assess the quality of English in this paper
( ) English very difficult to understand/incomprehensible
( ) Extensive editing of English language required
( ) Moderate editing of English language required
( ) Minor editing of English language required
(x) English language fine. No issues detected
|
Yes |
Can be improved |
Must be improved |
Not applicable |
|
|
Does the introduction provide sufficient background and include all relevant references? |
(x) |
( ) |
( ) |
( ) |
|
Are all the cited references relevant to the research? |
( ) |
(x) |
( ) |
( ) |
|
Is the research design appropriate? |
(x) |
( ) |
( ) |
( ) |
|
Are the methods adequately described? |
(x) |
( ) |
( ) |
( ) |
|
Are the results clearly presented? |
(x) |
( ) |
( ) |
( ) |
|
Are the conclusions supported by the results? |
(x) |
( ) |
( ) |
( ) |
Comments and Suggestions for Authors
The paper entitled ‘Ultra-sensitive determination of cadmium in food and water by flame-AAS after a new polyvinyl benzyl xanthate as an adsorbent based vortex assisted dispersive solid-phase microextraction: multivariate optimization’ reported a new polyvinyl benzyl xanthate (PvbXa) as adsorbent based vortex assisted dispersive solid-phase microextraction (VA-dSPµE) procedure was developed for the determination of cadmium (Cd) from food and water samples via flame atomic absorption spectrophotometry. This work in interesting. I have no specific concern but it is mandatory to add the manuscript in these point:
- In Figure 2, the interval between the horizontal coordinates of the chart are uneven. All the bonds in FTIR should be assigned.
The FTIR scales are accustomed standard scales, there comes from “Bruker FTIR instrument”. The characteristic bands were assigned as shown below:
- In Table 2, experiments design for factorial design, the Run 29 with a recovery at 97.0 %, however, subsequent research lacks corresponding validation.
Response: Thank you for valuable comments. We have redrawn the Table 2 and also checked their corresponding validation, hope now it will be acceptable.
Formun Üstü
Journal
Foods (ISSN 2304-8158)
Manuscript ID
foods-2578312
Type
Article
Title
Ultra-sensitive determination of cadmium in food and water by flame-AAS after a new polyvinyl benzyl xanthate as an adsorbent based vortex assisted dispersive solid-phase microextraction: multivariate optimization
Authors
Nail Altunay , Baki Hazer * , Muhammad Farooque Lanjwani , Mustafa Tuzen
Section
Food Analytical Methods
Abstract
A new polyvinyl benzyl xanthate (PvbXa) as adsorbent based vortex assisted dispersive solid-phase microextraction (VA-dSPµE) procedure was developed for the determination of cadmium (Cd) from food and water samples via flame atomic absorption spectrophotometry (FAAS). Synthesized PvbXa was characterized with 1H Nuclear magnetic resonance (NMR) Spectroscopy, Fourier Transform Infrared Spectroscopy (FTIR) and X-Ray Photoelectron Spectroscopy (XPS). The different parameters pH, sample volume, mixing type and time, sorbent amount, eluent time were optimized using standard analytical method. The optimized method for assessment of Cd in food and water samples shows good reliability. The good optimum conditions were found like 0.20–150 µg L-1 linear range, 0.06 µg L-1 LOD, 0.20 µg L-1 LOQ, 4.3 RSD %, and 200 preconcentration factor. The statistically experimental for variables were utilized using central composite design (CCD).
Formun Altı
Formun Üstü
Formun Üstü
Author's Reply to the Review Report (Reviewer 1)
Please provide a point-by-point response to the reviewer’s comments and either enter it in the box below or upload it as a Word/PDF file. Please write down "Please see the attachment." in the box if you only upload an attachment. An example can be found here.
* Author's Notes to Reviewer
FileEditViewInsertFormatToolsTableHelp
Paragraph
P
0 WORDS
Word / PDF
or
Formun Altı
Formun Üstü
Review Report Form
Open Review
(x) I would not like to sign my review report
( ) I would like to sign my review report
Quality of English Language
( ) I am not qualified to assess the quality of English in this paper
( ) English very difficult to understand/incomprehensible
( ) Extensive editing of English language required
( ) Moderate editing of English language required
( ) Minor editing of English language required
(x) English language fine. No issues detected
|
Yes |
Can be improved |
Must be improved |
Not applicable |
|
|
Does the introduction provide sufficient background and include all relevant references? |
(x) |
( ) |
( ) |
( ) |
|
Are all the cited references relevant to the research? |
( ) |
(x) |
( ) |
( ) |
|
Is the research design appropriate? |
(x) |
( ) |
( ) |
( ) |
|
Are the methods adequately described? |
(x) |
( ) |
( ) |
( ) |
|
Are the results clearly presented? |
(x) |
( ) |
( ) |
( ) |
|
Are the conclusions supported by the results? |
(x) |
( ) |
( ) |
( ) |
Comments and Suggestions for Authors
The paper entitled ‘Ultra-sensitive determination of cadmium in food and water by flame-AAS after a new polyvinyl benzyl xanthate as an adsorbent based vortex assisted dispersive solid-phase microextraction: multivariate optimization’ reported a new polyvinyl benzyl xanthate (PvbXa) as adsorbent based vortex assisted dispersive solid-phase microextraction (VA-dSPµE) procedure was developed for the determination of cadmium (Cd) from food and water samples via flame atomic absorption spectrophotometry. This work in interesting. I have no specific concern but it is mandatory to add the manuscript in these point:
- In Figure 2, the interval between the horizontal coordinates of the chart are uneven. All the bonds in FTIR should be assigned.
The FTIR scales are accustomed standard scales, there comes from “Bruker FTIR instrument”. The characteristic bands were assigned as shown below:
- In Table 2, experiments design for factorial design, the Run 29 with a recovery at 97.0 %, however, subsequent research lacks corresponding validation.
Response: Thank you for valuable comments. We have redrawn the Table 2 and also checked their corresponding validation, hope now it will be acceptable.
Formun Üstü
Journal
Foods (ISSN 2304-8158)
Manuscript ID
foods-2578312
Type
Article
Title
Ultra-sensitive determination of cadmium in food and water by flame-AAS after a new polyvinyl benzyl xanthate as an adsorbent based vortex assisted dispersive solid-phase microextraction: multivariate optimization
Authors
Nail Altunay , Baki Hazer * , Muhammad Farooque Lanjwani , Mustafa Tuzen
Section
Food Analytical Methods
Abstract
A new polyvinyl benzyl xanthate (PvbXa) as adsorbent based vortex assisted dispersive solid-phase microextraction (VA-dSPµE) procedure was developed for the determination of cadmium (Cd) from food and water samples via flame atomic absorption spectrophotometry (FAAS). Synthesized PvbXa was characterized with 1H Nuclear magnetic resonance (NMR) Spectroscopy, Fourier Transform Infrared Spectroscopy (FTIR) and X-Ray Photoelectron Spectroscopy (XPS). The different parameters pH, sample volume, mixing type and time, sorbent amount, eluent time were optimized using standard analytical method. The optimized method for assessment of Cd in food and water samples shows good reliability. The good optimum conditions were found like 0.20–150 µg L-1 linear range, 0.06 µg L-1 LOD, 0.20 µg L-1 LOQ, 4.3 RSD %, and 200 preconcentration factor. The statistically experimental for variables were utilized using central composite design (CCD).
Formun Altı
Formun Üstü
Formun Üstü
Author's Reply to the Review Report (Reviewer 1)
Please provide a point-by-point response to the reviewer’s comments and either enter it in the box below or upload it as a Word/PDF file. Please write down "Please see the attachment." in the box if you only upload an attachment. An example can be found here.
* Author's Notes to Reviewer
FileEditViewInsertFormatToolsTableHelp
Paragraph
P
0 WORDS
Word / PDF
or
Formun Altı
Formun Üstü
Review Report Form
Open Review
(x) I would not like to sign my review report
( ) I would like to sign my review report
Quality of English Language
( ) I am not qualified to assess the quality of English in this paper
( ) English very difficult to understand/incomprehensible
( ) Extensive editing of English language required
( ) Moderate editing of English language required
( ) Minor editing of English language required
(x) English language fine. No issues detected
|
Yes |
Can be improved |
Must be improved |
Not applicable |
|
|
Does the introduction provide sufficient background and include all relevant references? |
(x) |
( ) |
( ) |
( ) |
|
Are all the cited references relevant to the research? |
( ) |
(x) |
( ) |
( ) |
|
Is the research design appropriate? |
(x) |
( ) |
( ) |
( ) |
|
Are the methods adequately described? |
(x) |
( ) |
( ) |
( ) |
|
Are the results clearly presented? |
(x) |
( ) |
( ) |
( ) |
|
Are the conclusions supported by the results? |
(x) |
( ) |
( ) |
( ) |
Comments and Suggestions for Authors
The paper entitled ‘Ultra-sensitive determination of cadmium in food and water by flame-AAS after a new polyvinyl benzyl xanthate as an adsorbent based vortex assisted dispersive solid-phase microextraction: multivariate optimization’ reported a new polyvinyl benzyl xanthate (PvbXa) as adsorbent based vortex assisted dispersive solid-phase microextraction (VA-dSPµE) procedure was developed for the determination of cadmium (Cd) from food and water samples via flame atomic absorption spectrophotometry. This work in interesting. I have no specific concern but it is mandatory to add the manuscript in these point:
- In Figure 2, the interval between the horizontal coordinates of the chart are uneven. All the bonds in FTIR should be assigned.
The FTIR scales are accustomed standard scales, there comes from “Bruker FTIR instrument”. The characteristic bands were assigned as shown below:
- In Table 2, experiments design for factorial design, the Run 29 with a recovery at 97.0 %, however, subsequent research lacks corresponding validation.
Response: Thank you for valuable comments. We have redrawn the Table 2 and also checked their corresponding validation, hope now it will be acceptable.
Formun Üstü
Journal
Foods (ISSN 2304-8158)
Manuscript ID
foods-2578312
Type
Article
Title
Ultra-sensitive determination of cadmium in food and water by flame-AAS after a new polyvinyl benzyl xanthate as an adsorbent based vortex assisted dispersive solid-phase microextraction: multivariate optimization
Authors
Nail Altunay , Baki Hazer * , Muhammad Farooque Lanjwani , Mustafa Tuzen
Section
Food Analytical Methods
Abstract
A new polyvinyl benzyl xanthate (PvbXa) as adsorbent based vortex assisted dispersive solid-phase microextraction (VA-dSPµE) procedure was developed for the determination of cadmium (Cd) from food and water samples via flame atomic absorption spectrophotometry (FAAS). Synthesized PvbXa was characterized with 1H Nuclear magnetic resonance (NMR) Spectroscopy, Fourier Transform Infrared Spectroscopy (FTIR) and X-Ray Photoelectron Spectroscopy (XPS). The different parameters pH, sample volume, mixing type and time, sorbent amount, eluent time were optimized using standard analytical method. The optimized method for assessment of Cd in food and water samples shows good reliability. The good optimum conditions were found like 0.20–150 µg L-1 linear range, 0.06 µg L-1 LOD, 0.20 µg L-1 LOQ, 4.3 RSD %, and 200 preconcentration factor. The statistically experimental for variables were utilized using central composite design (CCD).
Formun Altı
Formun Üstü
Formun Üstü
Author's Reply to the Review Report (Reviewer 1)
Please provide a point-by-point response to the reviewer’s comments and either enter it in the box below or upload it as a Word/PDF file. Please write down "Please see the attachment." in the box if you only upload an attachment. An example can be found here.
* Author's Notes to Reviewer
FileEditViewInsertFormatToolsTableHelp
Paragraph
P
0 WORDS
Word / PDF
or
Formun Altı
Formun Üstü
Review Report Form
Open Review
(x) I would not like to sign my review report
( ) I would like to sign my review report
Quality of English Language
( ) I am not qualified to assess the quality of English in this paper
( ) English very difficult to understand/incomprehensible
( ) Extensive editing of English language required
( ) Moderate editing of English language required
( ) Minor editing of English language required
(x) English language fine. No issues detected
|
Yes |
Can be improved |
Must be improved |
Not applicable |
|
|
Does the introduction provide sufficient background and include all relevant references? |
(x) |
( ) |
( ) |
( ) |
|
Are all the cited references relevant to the research? |
( ) |
(x) |
( ) |
( ) |
|
Is the research design appropriate? |
(x) |
( ) |
( ) |
( ) |
|
Are the methods adequately described? |
(x) |
( ) |
( ) |
( ) |
|
Are the results clearly presented? |
(x) |
( ) |
( ) |
( ) |
|
Are the conclusions supported by the results? |
(x) |
( ) |
( ) |
( ) |
Comments and Suggestions for Authors
The paper entitled ‘Ultra-sensitive determination of cadmium in food and water by flame-AAS after a new polyvinyl benzyl xanthate as an adsorbent based vortex assisted dispersive solid-phase microextraction: multivariate optimization’ reported a new polyvinyl benzyl xanthate (PvbXa) as adsorbent based vortex assisted dispersive solid-phase microextraction (VA-dSPµE) procedure was developed for the determination of cadmium (Cd) from food and water samples via flame atomic absorption spectrophotometry. This work in interesting. I have no specific concern but it is mandatory to add the manuscript in these point:
- In Figure 2, the interval between the horizontal coordinates of the chart are uneven. All the bonds in FTIR should be assigned.
The FTIR scales are accustomed standard scales, there comes from “Bruker FTIR instrument”. The characteristic bands were assigned as shown below:
- In Table 2, experiments design for factorial design, the Run 29 with a recovery at 97.0 %, however, subsequent research lacks corresponding validation.
Response: Thank you for valuable comments. We have redrawn the Table 2 and also checked their corresponding validation, hope now it will be acceptable.
Formun Üstü
Journal
Foods (ISSN 2304-8158)
Manuscript ID
foods-2578312
Type
Article
Title
Ultra-sensitive determination of cadmium in food and water by flame-AAS after a new polyvinyl benzyl xanthate as an adsorbent based vortex assisted dispersive solid-phase microextraction: multivariate optimization
Authors
Nail Altunay , Baki Hazer * , Muhammad Farooque Lanjwani , Mustafa Tuzen
Section
Food Analytical Methods
Abstract
A new polyvinyl benzyl xanthate (PvbXa) as adsorbent based vortex assisted dispersive solid-phase microextraction (VA-dSPµE) procedure was developed for the determination of cadmium (Cd) from food and water samples via flame atomic absorption spectrophotometry (FAAS). Synthesized PvbXa was characterized with 1H Nuclear magnetic resonance (NMR) Spectroscopy, Fourier Transform Infrared Spectroscopy (FTIR) and X-Ray Photoelectron Spectroscopy (XPS). The different parameters pH, sample volume, mixing type and time, sorbent amount, eluent time were optimized using standard analytical method. The optimized method for assessment of Cd in food and water samples shows good reliability. The good optimum conditions were found like 0.20–150 µg L-1 linear range, 0.06 µg L-1 LOD, 0.20 µg L-1 LOQ, 4.3 RSD %, and 200 preconcentration factor. The statistically experimental for variables were utilized using central composite design (CCD).
Formun Altı
Formun Üstü
Formun Üstü
Author's Reply to the Review Report (Reviewer 1)
Please provide a point-by-point response to the reviewer’s comments and either enter it in the box below or upload it as a Word/PDF file. Please write down "Please see the attachment." in the box if you only upload an attachment. An example can be found here.
* Author's Notes to Reviewer
FileEditViewInsertFormatToolsTableHelp
Paragraph
P
0 WORDS
Word / PDF
or
Formun Altı
Formun Üstü
Review Report Form
Open Review
(x) I would not like to sign my review report
( ) I would like to sign my review report
Quality of English Language
( ) I am not qualified to assess the quality of English in this paper
( ) English very difficult to understand/incomprehensible
( ) Extensive editing of English language required
( ) Moderate editing of English language required
( ) Minor editing of English language required
(x) English language fine. No issues detected
|
Yes |
Can be improved |
Must be improved |
Not applicable |
|
|
Does the introduction provide sufficient background and include all relevant references? |
(x) |
( ) |
( ) |
( ) |
|
Are all the cited references relevant to the research? |
( ) |
(x) |
( ) |
( ) |
|
Is the research design appropriate? |
(x) |
( ) |
( ) |
( ) |
|
Are the methods adequately described? |
(x) |
( ) |
( ) |
( ) |
|
Are the results clearly presented? |
(x) |
( ) |
( ) |
( ) |
|
Are the conclusions supported by the results? |
(x) |
( ) |
( ) |
( ) |
Comments and Suggestions for Authors
The paper entitled ‘Ultra-sensitive determination of cadmium in food and water by flame-AAS after a new polyvinyl benzyl xanthate as an adsorbent based vortex assisted dispersive solid-phase microextraction: multivariate optimization’ reported a new polyvinyl benzyl xanthate (PvbXa) as adsorbent based vortex assisted dispersive solid-phase microextraction (VA-dSPµE) procedure was developed for the determination of cadmium (Cd) from food and water samples via flame atomic absorption spectrophotometry. This work in interesting. I have no specific concern but it is mandatory to add the manuscript in these point:
- In Figure 2, the interval between the horizontal coordinates of the chart are uneven. All the bonds in FTIR should be assigned.
The FTIR scales are accustomed standard scales, there comes from “Bruker FTIR instrument”. The characteristic bands were assigned as shown below:
- In Table 2, experiments design for factorial design, the Run 29 with a recovery at 97.0 %, however, subsequent research lacks corresponding validation.
Response: Thank you for valuable comments. We have redrawn the Table 2 and also checked their corresponding validation, hope now it will be acceptable.
Formun Üstü
Journal
Foods (ISSN 2304-8158)
Manuscript ID
foods-2578312
Type
Article
Title
Ultra-sensitive determination of cadmium in food and water by flame-AAS after a new polyvinyl benzyl xanthate as an adsorbent based vortex assisted dispersive solid-phase microextraction: multivariate optimization
Authors
Nail Altunay , Baki Hazer * , Muhammad Farooque Lanjwani , Mustafa Tuzen
Section
Food Analytical Methods
Abstract
A new polyvinyl benzyl xanthate (PvbXa) as adsorbent based vortex assisted dispersive solid-phase microextraction (VA-dSPµE) procedure was developed for the determination of cadmium (Cd) from food and water samples via flame atomic absorption spectrophotometry (FAAS). Synthesized PvbXa was characterized with 1H Nuclear magnetic resonance (NMR) Spectroscopy, Fourier Transform Infrared Spectroscopy (FTIR) and X-Ray Photoelectron Spectroscopy (XPS). The different parameters pH, sample volume, mixing type and time, sorbent amount, eluent time were optimized using standard analytical method. The optimized method for assessment of Cd in food and water samples shows good reliability. The good optimum conditions were found like 0.20–150 µg L-1 linear range, 0.06 µg L-1 LOD, 0.20 µg L-1 LOQ, 4.3 RSD %, and 200 preconcentration factor. The statistically experimental for variables were utilized using central composite design (CCD).
Formun Altı
Formun Üstü
Formun Üstü
Author's Reply to the Review Report (Reviewer 1)
Please provide a point-by-point response to the reviewer’s comments and either enter it in the box below or upload it as a Word/PDF file. Please write down "Please see the attachment." in the box if you only upload an attachment. An example can be found here.
* Author's Notes to Reviewer
FileEditViewInsertFormatToolsTableHelp
Paragraph
P
0 WORDS
Word / PDF
or
Formun Altı
Formun Üstü
Review Report Form
Open Review
(x) I would not like to sign my review report
( ) I would like to sign my review report
Quality of English Language
( ) I am not qualified to assess the quality of English in this paper
( ) English very difficult to understand/incomprehensible
( ) Extensive editing of English language required
( ) Moderate editing of English language required
( ) Minor editing of English language required
(x) English language fine. No issues detected
|
Yes |
Can be improved |
Must be improved |
Not applicable |
|
|
Does the introduction provide sufficient background and include all relevant references? |
(x) |
( ) |
( ) |
( ) |
|
Are all the cited references relevant to the research? |
( ) |
(x) |
( ) |
( ) |
|
Is the research design appropriate? |
(x) |
( ) |
( ) |
( ) |
|
Are the methods adequately described? |
(x) |
( ) |
( ) |
( ) |
|
Are the results clearly presented? |
(x) |
( ) |
( ) |
( ) |
|
Are the conclusions supported by the results? |
(x) |
( ) |
( ) |
( ) |
Comments and Suggestions for Authors
The paper entitled ‘Ultra-sensitive determination of cadmium in food and water by flame-AAS after a new polyvinyl benzyl xanthate as an adsorbent based vortex assisted dispersive solid-phase microextraction: multivariate optimization’ reported a new polyvinyl benzyl xanthate (PvbXa) as adsorbent based vortex assisted dispersive solid-phase microextraction (VA-dSPµE) procedure was developed for the determination of cadmium (Cd) from food and water samples via flame atomic absorption spectrophotometry. This work in interesting. I have no specific concern but it is mandatory to add the manuscript in these point:
- In Figure 2, the interval between the horizontal coordinates of the chart are uneven. All the bonds in FTIR should be assigned.
The FTIR scales are accustomed standard scales, there comes from “Bruker FTIR instrument”. The characteristic bands were assigned as shown below:
- In Table 2, experiments design for factorial design, the Run 29 with a recovery at 97.0 %, however, subsequent research lacks corresponding validation.
Response: Thank you for valuable comments. We have redrawn the Table 2 and also checked their corresponding validation, hope now it will be acceptable.
Formun Üstü
Journal
Foods (ISSN 2304-8158)
Manuscript ID
foods-2578312
Type
Article
Title
Ultra-sensitive determination of cadmium in food and water by flame-AAS after a new polyvinyl benzyl xanthate as an adsorbent based vortex assisted dispersive solid-phase microextraction: multivariate optimization
Authors
Nail Altunay , Baki Hazer * , Muhammad Farooque Lanjwani , Mustafa Tuzen
Section
Food Analytical Methods
Abstract
A new polyvinyl benzyl xanthate (PvbXa) as adsorbent based vortex assisted dispersive solid-phase microextraction (VA-dSPµE) procedure was developed for the determination of cadmium (Cd) from food and water samples via flame atomic absorption spectrophotometry (FAAS). Synthesized PvbXa was characterized with 1H Nuclear magnetic resonance (NMR) Spectroscopy, Fourier Transform Infrared Spectroscopy (FTIR) and X-Ray Photoelectron Spectroscopy (XPS). The different parameters pH, sample volume, mixing type and time, sorbent amount, eluent time were optimized using standard analytical method. The optimized method for assessment of Cd in food and water samples shows good reliability. The good optimum conditions were found like 0.20–150 µg L-1 linear range, 0.06 µg L-1 LOD, 0.20 µg L-1 LOQ, 4.3 RSD %, and 200 preconcentration factor. The statistically experimental for variables were utilized using central composite design (CCD).
Formun Altı
Formun Üstü
Formun Üstü
Author's Reply to the Review Report (Reviewer 1)
Please provide a point-by-point response to the reviewer’s comments and either enter it in the box below or upload it as a Word/PDF file. Please write down "Please see the attachment." in the box if you only upload an attachment. An example can be found here.
* Author's Notes to Reviewer
FileEditViewInsertFormatToolsTableHelp
Paragraph
P
0 WORDS
Word / PDF
or
Formun Altı
Formun Üstü
Review Report Form
Open Review
(x) I would not like to sign my review report
( ) I would like to sign my review report
Quality of English Language
( ) I am not qualified to assess the quality of English in this paper
( ) English very difficult to understand/incomprehensible
( ) Extensive editing of English language required
( ) Moderate editing of English language required
( ) Minor editing of English language required
(x) English language fine. No issues detected
|
Yes |
Can be improved |
Must be improved |
Not applicable |
|
|
Does the introduction provide sufficient background and include all relevant references? |
(x) |
( ) |
( ) |
( ) |
|
Are all the cited references relevant to the research? |
( ) |
(x) |
( ) |
( ) |
|
Is the research design appropriate? |
(x) |
( ) |
( ) |
( ) |
|
Are the methods adequately described? |
(x) |
( ) |
( ) |
( ) |
|
Are the results clearly presented? |
(x) |
( ) |
( ) |
( ) |
|
Are the conclusions supported by the results? |
(x) |
( ) |
( ) |
( ) |
Comments and Suggestions for Authors
The paper entitled ‘Ultra-sensitive determination of cadmium in food and water by flame-AAS after a new polyvinyl benzyl xanthate as an adsorbent based vortex assisted dispersive solid-phase microextraction: multivariate optimization’ reported a new polyvinyl benzyl xanthate (PvbXa) as adsorbent based vortex assisted dispersive solid-phase microextraction (VA-dSPµE) procedure was developed for the determination of cadmium (Cd) from food and water samples via flame atomic absorption spectrophotometry. This work in interesting. I have no specific concern but it is mandatory to add the manuscript in these point:
- In Figure 2, the interval between the horizontal coordinates of the chart are uneven. All the bonds in FTIR should be assigned.
The FTIR scales are accustomed standard scales, there comes from “Bruker FTIR instrument”. The characteristic bands were assigned as shown below:
- In Table 2, experiments design for factorial design, the Run 29 with a recovery at 97.0 %, however, subsequent research lacks corresponding validation.
Response: Thank you for valuable comments. We have redrawn the Table 2 and also checked their corresponding validation, hope now it will be acceptable.
Formun Üstü
Journal
Foods (ISSN 2304-8158)
Manuscript ID
foods-2578312
Type
Article
Title
Ultra-sensitive determination of cadmium in food and water by flame-AAS after a new polyvinyl benzyl xanthate as an adsorbent based vortex assisted dispersive solid-phase microextraction: multivariate optimization
Authors
Nail Altunay , Baki Hazer * , Muhammad Farooque Lanjwani , Mustafa Tuzen
Section
Food Analytical Methods
Abstract
A new polyvinyl benzyl xanthate (PvbXa) as adsorbent based vortex assisted dispersive solid-phase microextraction (VA-dSPµE) procedure was developed for the determination of cadmium (Cd) from food and water samples via flame atomic absorption spectrophotometry (FAAS). Synthesized PvbXa was characterized with 1H Nuclear magnetic resonance (NMR) Spectroscopy, Fourier Transform Infrared Spectroscopy (FTIR) and X-Ray Photoelectron Spectroscopy (XPS). The different parameters pH, sample volume, mixing type and time, sorbent amount, eluent time were optimized using standard analytical method. The optimized method for assessment of Cd in food and water samples shows good reliability. The good optimum conditions were found like 0.20–150 µg L-1 linear range, 0.06 µg L-1 LOD, 0.20 µg L-1 LOQ, 4.3 RSD %, and 200 preconcentration factor. The statistically experimental for variables were utilized using central composite design (CCD).
Formun Altı
Formun Üstü
Formun Üstü
Author's Reply to the Review Report (Reviewer 1)
Please provide a point-by-point response to the reviewer’s comments and either enter it in the box below or upload it as a Word/PDF file. Please write down "Please see the attachment." in the box if you only upload an attachment. An example can be found here.
* Author's Notes to Reviewer
FileEditViewInsertFormatToolsTableHelp
Paragraph
P
0 WORDS
Word / PDF
or
Formun Altı
Formun Üstü
Review Report Form
Open Review
(x) I would not like to sign my review report
( ) I would like to sign my review report
Quality of English Language
( ) I am not qualified to assess the quality of English in this paper
( ) English very difficult to understand/incomprehensible
( ) Extensive editing of English language required
( ) Moderate editing of English language required
( ) Minor editing of English language required
(x) English language fine. No issues detected
|
Yes |
Can be improved |
Must be improved |
Not applicable |
|
|
Does the introduction provide sufficient background and include all relevant references? |
(x) |
( ) |
( ) |
( ) |
|
Are all the cited references relevant to the research? |
( ) |
(x) |
( ) |
( ) |
|
Is the research design appropriate? |
(x) |
( ) |
( ) |
( ) |
|
Are the methods adequately described? |
(x) |
( ) |
( ) |
( ) |
|
Are the results clearly presented? |
(x) |
( ) |
( ) |
( ) |
|
Are the conclusions supported by the results? |
(x) |
( ) |
( ) |
( ) |
Comments and Suggestions for Authors
The paper entitled ‘Ultra-sensitive determination of cadmium in food and water by flame-AAS after a new polyvinyl benzyl xanthate as an adsorbent based vortex assisted dispersive solid-phase microextraction: multivariate optimization’ reported a new polyvinyl benzyl xanthate (PvbXa) as adsorbent based vortex assisted dispersive solid-phase microextraction (VA-dSPµE) procedure was developed for the determination of cadmium (Cd) from food and water samples via flame atomic absorption spectrophotometry. This work in interesting. I have no specific concern but it is mandatory to add the manuscript in these point:
- In Figure 2, the interval between the horizontal coordinates of the chart are uneven. All the bonds in FTIR should be assigned.
The FTIR scales are accustomed standard scales, there comes from “Bruker FTIR instrument”. The characteristic bands were assigned as shown below:
- In Table 2, experiments design for factorial design, the Run 29 with a recovery at 97.0 %, however, subsequent research lacks corresponding validation.
Response: Thank you for valuable comments. We have redrawn the Table 2 and also checked their corresponding validation, hope now it will be acceptable.
Formun Üstü
Journal
Foods (ISSN 2304-8158)
Manuscript ID
foods-2578312
Type
Article
Title
Ultra-sensitive determination of cadmium in food and water by flame-AAS after a new polyvinyl benzyl xanthate as an adsorbent based vortex assisted dispersive solid-phase microextraction: multivariate optimization
Authors
Nail Altunay , Baki Hazer * , Muhammad Farooque Lanjwani , Mustafa Tuzen
Section
Food Analytical Methods
Abstract
A new polyvinyl benzyl xanthate (PvbXa) as adsorbent based vortex assisted dispersive solid-phase microextraction (VA-dSPµE) procedure was developed for the determination of cadmium (Cd) from food and water samples via flame atomic absorption spectrophotometry (FAAS). Synthesized PvbXa was characterized with 1H Nuclear magnetic resonance (NMR) Spectroscopy, Fourier Transform Infrared Spectroscopy (FTIR) and X-Ray Photoelectron Spectroscopy (XPS). The different parameters pH, sample volume, mixing type and time, sorbent amount, eluent time were optimized using standard analytical method. The optimized method for assessment of Cd in food and water samples shows good reliability. The good optimum conditions were found like 0.20–150 µg L-1 linear range, 0.06 µg L-1 LOD, 0.20 µg L-1 LOQ, 4.3 RSD %, and 200 preconcentration factor. The statistically experimental for variables were utilized using central composite design (CCD).
Formun Altı
Formun Üstü
Formun Üstü
Author's Reply to the Review Report (Reviewer 1)
Please provide a point-by-point response to the reviewer’s comments and either enter it in the box below or upload it as a Word/PDF file. Please write down "Please see the attachment." in the box if you only upload an attachment. An example can be found here.
* Author's Notes to Reviewer
FileEditViewInsertFormatToolsTableHelp
Paragraph
P
0 WORDS
Word / PDF
or
Formun Altı
Formun Üstü
Review Report Form
Open Review
(x) I would not like to sign my review report
( ) I would like to sign my review report
Quality of English Language
( ) I am not qualified to assess the quality of English in this paper
( ) English very difficult to understand/incomprehensible
( ) Extensive editing of English language required
( ) Moderate editing of English language required
( ) Minor editing of English language required
(x) English language fine. No issues detected
|
Yes |
Can be improved |
Must be improved |
Not applicable |
|
|
Does the introduction provide sufficient background and include all relevant references? |
(x) |
( ) |
( ) |
( ) |
|
Are all the cited references relevant to the research? |
( ) |
(x) |
( ) |
( ) |
|
Is the research design appropriate? |
(x) |
( ) |
( ) |
( ) |
|
Are the methods adequately described? |
(x) |
( ) |
( ) |
( ) |
|
Are the results clearly presented? |
(x) |
( ) |
( ) |
( ) |
|
Are the conclusions supported by the results? |
(x) |
( ) |
( ) |
( ) |
Comments and Suggestions for Authors
The paper entitled ‘Ultra-sensitive determination of cadmium in food and water by flame-AAS after a new polyvinyl benzyl xanthate as an adsorbent based vortex assisted dispersive solid-phase microextraction: multivariate optimization’ reported a new polyvinyl benzyl xanthate (PvbXa) as adsorbent based vortex assisted dispersive solid-phase microextraction (VA-dSPµE) procedure was developed for the determination of cadmium (Cd) from food and water samples via flame atomic absorption spectrophotometry. This work in interesting. I have no specific concern but it is mandatory to add the manuscript in these point:
- In Figure 2, the interval between the horizontal coordinates of the chart are uneven. All the bonds in FTIR should be assigned.
The FTIR scales are accustomed standard scales, there comes from “Bruker FTIR instrument”. The characteristic bands were assigned as shown below:
- In Table 2, experiments design for factorial design, the Run 29 with a recovery at 97.0 %, however, subsequent research lacks corresponding validation.
Response: Thank you for valuable comments. We have redrawn the Table 2 and also checked their corresponding validation, hope now it will be acceptable.
Formun Üstü
Journal
Foods (ISSN 2304-8158)
Manuscript ID
foods-2578312
Type
Article
Title
Ultra-sensitive determination of cadmium in food and water by flame-AAS after a new polyvinyl benzyl xanthate as an adsorbent based vortex assisted dispersive solid-phase microextraction: multivariate optimization
Authors
Nail Altunay , Baki Hazer * , Muhammad Farooque Lanjwani , Mustafa Tuzen
Section
Food Analytical Methods
Abstract
A new polyvinyl benzyl xanthate (PvbXa) as adsorbent based vortex assisted dispersive solid-phase microextraction (VA-dSPµE) procedure was developed for the determination of cadmium (Cd) from food and water samples via flame atomic absorption spectrophotometry (FAAS). Synthesized PvbXa was characterized with 1H Nuclear magnetic resonance (NMR) Spectroscopy, Fourier Transform Infrared Spectroscopy (FTIR) and X-Ray Photoelectron Spectroscopy (XPS). The different parameters pH, sample volume, mixing type and time, sorbent amount, eluent time were optimized using standard analytical method. The optimized method for assessment of Cd in food and water samples shows good reliability. The good optimum conditions were found like 0.20–150 µg L-1 linear range, 0.06 µg L-1 LOD, 0.20 µg L-1 LOQ, 4.3 RSD %, and 200 preconcentration factor. The statistically experimental for variables were utilized using central composite design (CCD).
Formun Altı
Formun Üstü
Formun Üstü
Author's Reply to the Review Report (Reviewer 1)
Please provide a point-by-point response to the reviewer’s comments and either enter it in the box below or upload it as a Word/PDF file. Please write down "Please see the attachment." in the box if you only upload an attachment. An example can be found here.
* Author's Notes to Reviewer
FileEditViewInsertFormatToolsTableHelp
Paragraph
P
0 WORDS
Word / PDF
or
Formun Altı
Formun Üstü
Review Report Form
Open Review
(x) I would not like to sign my review report
( ) I would like to sign my review report
Quality of English Language
( ) I am not qualified to assess the quality of English in this paper
( ) English very difficult to understand/incomprehensible
( ) Extensive editing of English language required
( ) Moderate editing of English language required
( ) Minor editing of English language required
(x) English language fine. No issues detected
|
Yes |
Can be improved |
Must be improved |
Not applicable |
|
|
Does the introduction provide sufficient background and include all relevant references? |
(x) |
( ) |
( ) |
( ) |
|
Are all the cited references relevant to the research? |
( ) |
(x) |
( ) |
( ) |
|
Is the research design appropriate? |
(x) |
( ) |
( ) |
( ) |
|
Are the methods adequately described? |
(x) |
( ) |
( ) |
( ) |
|
Are the results clearly presented? |
(x) |
( ) |
( ) |
( ) |
|
Are the conclusions supported by the results? |
(x) |
( ) |
( ) |
( ) |
Comments and Suggestions for Authors
The paper entitled ‘Ultra-sensitive determination of cadmium in food and water by flame-AAS after a new polyvinyl benzyl xanthate as an adsorbent based vortex assisted dispersive solid-phase microextraction: multivariate optimization’ reported a new polyvinyl benzyl xanthate (PvbXa) as adsorbent based vortex assisted dispersive solid-phase microextraction (VA-dSPµE) procedure was developed for the determination of cadmium (Cd) from food and water samples via flame atomic absorption spectrophotometry. This work in interesting. I have no specific concern but it is mandatory to add the manuscript in these point:
- In Figure 2, the interval between the horizontal coordinates of the chart are uneven. All the bonds in FTIR should be assigned.
The FTIR scales are accustomed standard scales, there comes from “Bruker FTIR instrument”. The characteristic bands were assigned as shown below:
- In Table 2, experiments design for factorial design, the Run 29 with a recovery at 97.0 %, however, subsequent research lacks corresponding validation.
Response: Thank you for valuable comments. We have redrawn the Table 2 and also checked their corresponding validation, hope now it will be acceptable.
Formun Üstü
Journal
Foods (ISSN 2304-8158)
Manuscript ID
foods-2578312
Type
Article
Title
Ultra-sensitive determination of cadmium in food and water by flame-AAS after a new polyvinyl benzyl xanthate as an adsorbent based vortex assisted dispersive solid-phase microextraction: multivariate optimization
Authors
Nail Altunay , Baki Hazer * , Muhammad Farooque Lanjwani , Mustafa Tuzen
Section
Food Analytical Methods
Abstract
A new polyvinyl benzyl xanthate (PvbXa) as adsorbent based vortex assisted dispersive solid-phase microextraction (VA-dSPµE) procedure was developed for the determination of cadmium (Cd) from food and water samples via flame atomic absorption spectrophotometry (FAAS). Synthesized PvbXa was characterized with 1H Nuclear magnetic resonance (NMR) Spectroscopy, Fourier Transform Infrared Spectroscopy (FTIR) and X-Ray Photoelectron Spectroscopy (XPS). The different parameters pH, sample volume, mixing type and time, sorbent amount, eluent time were optimized using standard analytical method. The optimized method for assessment of Cd in food and water samples shows good reliability. The good optimum conditions were found like 0.20–150 µg L-1 linear range, 0.06 µg L-1 LOD, 0.20 µg L-1 LOQ, 4.3 RSD %, and 200 preconcentration factor. The statistically experimental for variables were utilized using central composite design (CCD).
Formun Altı
Formun Üstü
Reviewer 2 Report
This is nicely written paper recommended for the publication
satisfactory
Author Response
Author's Reply to the Review Report (Reviewer 2)
Please provide a point-by-point response to the reviewer’s comments and either enter it in the box below or upload it as a Word/PDF file. Please write down "Please see the attachment." in the box if you only upload an attachment. An example can be found here.
* Author's Notes to Reviewer
FileEditViewInsertFormatToolsTableHelp
Paragraph
P
0 WORDS
Word / PDF
or
Formun Altı
Formun Üstü
Review Report Form
Open Review
( ) I would not like to sign my review report
(x) I would like to sign my review report
Quality of English Language
( ) I am not qualified to assess the quality of English in this paper
( ) English very difficult to understand/incomprehensible
( ) Extensive editing of English language required
(x) Moderate editing of English language required
( ) Minor editing of English language required
( ) English language fine. No issues detected
|
Yes |
Can be improved |
Must be improved |
Not applicable |
|
|
Does the introduction provide sufficient background and include all relevant references? |
(x) |
( ) |
( ) |
( ) |
|
Are all the cited references relevant to the research? |
(x) |
( ) |
( ) |
( ) |
|
Is the research design appropriate? |
(x) |
( ) |
( ) |
( ) |
|
Are the methods adequately described? |
(x) |
( ) |
( ) |
( ) |
|
Are the results clearly presented? |
(x) |
( ) |
( ) |
( ) |
|
Are the conclusions supported by the results? |
(x) |
( ) |
( ) |
( ) |
Comments and Suggestions for Authors
This is nicely written paper recommended for the publication
Comments on the Quality of English Language
satisfactory
Submission Date
10 August 2023
Date of this review
21 Aug 2023 09:00:43
Formun Altı
Formun Üstü
Journal
Foods (ISSN 2304-8158)
Manuscript ID
foods-2578312
Type
Article
Title
Ultra-sensitive determination of cadmium in food and water by flame-AAS after a new polyvinyl benzyl xanthate as an adsorbent based vortex assisted dispersive solid-phase microextraction: multivariate optimization
Authors
Nail Altunay , Baki Hazer * , Muhammad Farooque Lanjwani , Mustafa Tuzen
Section
Food Analytical Methods
Abstract
A new polyvinyl benzyl xanthate (PvbXa) as adsorbent based vortex assisted dispersive solid-phase microextraction (VA-dSPµE) procedure was developed for the determination of cadmium (Cd) from food and water samples via flame atomic absorption spectrophotometry (FAAS). Synthesized PvbXa was characterized with 1H Nuclear magnetic resonance (NMR) Spectroscopy, Fourier Transform Infrared Spectroscopy (FTIR) and X-Ray Photoelectron Spectroscopy (XPS). The different parameters pH, sample volume, mixing type and time, sorbent amount, eluent time were optimized using standard analytical method. The optimized method for assessment of Cd in food and water samples shows good reliability. The good optimum conditions were found like 0.20–150 µg L-1 linear range, 0.06 µg L-1 LOD, 0.20 µg L-1 LOQ, 4.3 RSD %, and 200 preconcentration factor. The statistically experimental for variables were utilized using central composite design (CCD).
Formun Altı
Formun Üstü
Reviewer 3 Report
The manuscript presents a new polyvinyl benzyl xanthate (PvbXa) as adsorbent based vortex assisted dispersive solid-phase microextraction (VAdSPµE) procedure for the determination of cadmium (Cd) from food and water samples via flame atomic absorption spectrophotometry (FAAS).
Although the paper is well designed and described it is hard to say that it fits into the scope of this journal even if there is a section in the scope and aims devoted to Food analysis.
I would suggest the authors to send the manuscript to a journal with a more technological or analytical profile.
No results on te Cd levels or risks are communicated from this paper.
Author Response
Author's Reply to the Review Report (Reviewer 3)
Please provide a point-by-point response to the reviewer’s comments and either enter it in the box below or upload it as a Word/PDF file. Please write down "Please see the attachment." in the box if you only upload an attachment. An example can be found here.
* Author's Notes to Reviewer
FileEditViewInsertFormatToolsTableHelp
Paragraph
P
0 WORDS
Word / PDF
or
Formun Altı
Formun Üstü
Review Report Form
Open Review
( ) I would not like to sign my review report
(x) I would like to sign my review report
Quality of English Language
( ) I am not qualified to assess the quality of English in this paper
( ) English very difficult to understand/incomprehensible
( ) Extensive editing of English language required
( ) Moderate editing of English language required
( ) Minor editing of English language required
(x) English language fine. No issues detected
|
Yes |
Can be improved |
Must be improved |
Not applicable |
|
|
Does the introduction provide sufficient background and include all relevant references? |
( ) |
(x) |
( ) |
( ) |
|
Are all the cited references relevant to the research? |
(x) |
( ) |
( ) |
( ) |
|
Is the research design appropriate? |
(x) |
( ) |
( ) |
( ) |
|
Are the methods adequately described? |
(x) |
( ) |
( ) |
( ) |
|
Are the results clearly presented? |
(x) |
( ) |
( ) |
( ) |
|
Are the conclusions supported by the results? |
( ) |
(x) |
( ) |
( ) |
Comments and Suggestions for Authors
The manuscript presents a new polyvinyl benzyl xanthate (PvbXa) as adsorbent based vortex assisted dispersive solid-phase microextraction (VAdSPµE) procedure for the determination of cadmium (Cd) from food and water samples via flame atomic absorption spectrophotometry (FAAS).
Although the paper is well designed and described it is hard to say that it fits into the scope of this journal even if there is a section in the scope and aims devoted to Food analysis.
I would suggest the authors to send the manuscript to a journal with a more technological or analytical profile.
No results on te Cd levels or risks are communicated from this paper.
Response: Cadmium levels in the measured samples were found in µg g-1 level with 1.4-4.3% RSD value (Table 5b) and achieved result was found in WHO limit [14]. There are no any risks to consumption of analyzed food samples with respect to human health. However, food samples should be more often analysis with respect to cadmium.
Submission Date
10 August 2023
Date of this review
29 Aug 2023 12:57:17
Formun Altı
Formun Üstü
Journal
Foods (ISSN 2304-8158)
Manuscript ID
foods-2578312
Type
Article
Title
Ultra-sensitive determination of cadmium in food and water by flame-AAS after a new polyvinyl benzyl xanthate as an adsorbent based vortex assisted dispersive solid-phase microextraction: multivariate optimization
Authors
Nail Altunay , Baki Hazer * , Muhammad Farooque Lanjwani , Mustafa Tuzen
Section
Food Analytical Methods
Abstract
A new polyvinyl benzyl xanthate (PvbXa) as adsorbent based vortex assisted dispersive solid-phase microextraction (VA-dSPµE) procedure was developed for the determination of cadmium (Cd) from food and water samples via flame atomic absorption spectrophotometry (FAAS). Synthesized PvbXa was characterized with 1H Nuclear magnetic resonance (NMR) Spectroscopy, Fourier Transform Infrared Spectroscopy (FTIR) and X-Ray Photoelectron Spectroscopy (XPS). The different parameters pH, sample volume, mixing type and time, sorbent amount, eluent time were optimized using standard analytical method. The optimized method for assessment of Cd in food and water samples shows good reliability. The good optimum conditions were found like 0.20–150 µg L-1 linear range, 0.06 µg L-1 LOD, 0.20 µg L-1 LOQ, 4.3 RSD %, and 200 preconcentration factor. The statistically experimental for variables were utilized using central composite design (CCD).
Formun Altı
Formun Üstü
Reviewer 4 Report
A valuable piece of work with numerous interesting results can be observed in your text; nevertheless, I do have some comments and questions.
The abbreviation for cadmium is mentioned twice in the abstract: once in the first line and again in the fourth line. This repetition is unnecessary.
Why is the term “spectrophotometry” specifically used for FAAS, while “spectroscopy” or “spectrometry” is used for other spectrometric methods? Additionally, other name such as “flame atomic absorption spectrophotometer” for FAAS can be also found on page 2, paragraph 2, line 2.
Sentence: “The FAAS showed great consideration for assess and extraction of metals because of low cost, easy to operate, fast response and best precision and accuracy.” (page 2, paragraph 2) I understand your point, but FAAS doesn't prioritize the thorough extraction of metals. It primarily serves as a quantification method, finding utility in trace analysis when paired with an efficient separation/preconcentration technique for the target analyte.
Reference number 15 is duplicated: once as a numeral and another time as “(Xing et al., 2020)” (page 2, paragraph 1, line 15).
Sentence: “The low detection limit of FAAS and matrix constituents of real samples, it is necessary to use preconcentration techniques, solid phase extraction, dispersive liquid liquid microextraction, etc. [24-26].” Detection limit of FAAS is relatively high. I think you wanted to write: The absence of a low detection limit and the presence of various co-existing matrix components are the two main reasons why employing an effective preconcentration technique becomes necessary for reliable quantification of trace elements using FAAS.
While the abbreviation of VA-dSPµE is explained in the abstract, there is no explanation provided when it is first used in the main text (page 2, paragraph 4, line 2).
In the introduction, nearly half of the text is devoted to rice, with no mention of cadmium in other foodstuffs. Don't we have any information about cadmium in other types of food? Your interest wasn't limited to just rice samples; you also examined apple, spinach, salad, tomatoes, onion, oat, corn, aubergine, wheat, and mushroom samples.
I believe that the standout aspect of your work is the utilization of a novel sorbent for cadmium, namely PvbXa. While the introduction merely contains a single sentence about this significant development, it lacks an elaboration on the theory or hypothesis behind the sorption mechanism between PvbXa and Cd. This absence of discussion could be particularly intriguing, especially considering the primary focus of your work.
Part 2.4.
Sentence: “Collected water samples were first filtered using a micro filter into a 250 mL beaker.” What is the size pore of the filter you used? No acid was added to your water samples after collection, right? (just for stabilization)
Sentence: “It was then digested via evaporation on the heating plate until the final volume was 10 mL.” From this, I interpreted it as a simple evaporation process. However, does this imply that the water samples were preconcentrated 25 times in this step (from 250 mL to 10 mL)? I couldn't find any information about this preconcentration in the main text.
Sentence: “Collected food samples were first homogenized by pulverizing with a lab grinder.” Did all your actual solid samples undergo the same process? Were apple, spinach, salad, tomatoes, onion, aubergine, and mushroom samples not dried before being subjected to this homogenization process?
Sentence: “Then, a 1:4 mixture of hydrogen peroxide and nitric acid was added to the samples.” What volume of this mixture was used?
Part 2.6.
Text: “To get the Cd(II) ions adsorbed on the solid back into the measurement solution, 1250 µL of EtOH was added and vortexed for 120 s. Finally, Cd determination was performed by injecting the death solution into the atomization section of the FAAS.” Does this mean that the sorbent was dissolved? If elution was performed, another separation step should be included in the subsequent procedure after shaking the sorbent with the eluting agent, such as centrifugation or filtration.
Part 2.7.
Sentence: “The 30 experimental run drawn in the factorial design used 4 variables pH, sorbent amount mg, mixing time (min) and sample volume mL Table 2.” For consistency, maybe (mg) and (mL) should also be in parentheses. Maybe something like “it can be seen in Table 2” can be incorporated to this sentence.
Part 3.2.
Sentence: “Therefore, EtOH volume 1200 µL was adjusted for study.” In section 2.6., a volume of 1250 µL is mentioned. Which value is correct?
Part 3.3.
Sentence: “Present VA-dSPµE method has highly selective for uses to water and environmental samples for extraction the Cd ions.” Maybe “food samples” can be also mentioned in this sentence.
Parts 3.4. and 3.5.
In the main text, “µg L-1” is used as the concentration unit, while in tables 3, 4, and 5a, “ng mL-1” is used. Please use the same unit for this parameter throughout the entire article.
While the abbreviation “VA-dSPµE” is used in the text, the abbreviation “VA-SPME” is used in the titles of tables 3, 4, 5a, and 5b. Please use the same abbreviation throughout the entire article.
Text: “Intraday precision (N=5) was found 2.4, 3.1 and 3.6% for 1, 50 and 100 µg L-1 of Cd(II), respectively. Inter-day precision (N=5) was found 2.7, 3.7 and 4.3% for 1, 50 and 100 µg L-1 of Cd(II).” This trend of RSD values is unusual. Typically, when analyzing higher concentrations, lower RSD values can be achieved.
Sentence: “The PF, EF and ER of VA-dSPµE method was achieve 200, 100, 97.1% respectively (see Table 3).” How were calculated the PF and EF values? Sometimes, published literature can contain confusing information about these parameters. An explanation could help more readers understand the difference between these values.
Table 5.a.
I'm having trouble understanding how the spiked samples were prepared and measured. Is the obtained concentration value (for Cd in original sample without the spike) valid for your samples after evaporation (from 250 mL to 10 mL)? Was 75 µg L-1 of Cd added before the extraction procedure? Then, was the sample preconcentrated 200 times? Finally, was a measured concentration of around 15,000 µg L-1 achieved? Is the linear range 0.20 – 150 µg L-1? Additionally, it seems that even Cd in the original sample (without the spike) can be preconcentrated 200 times after using the extraction procedure. I'm quite confused.
Part. 4.
Sentence: “The value of pH is a critical factor that may significantly affect the efficiency of extraction and metal separation through microextraction procedure comprises prior formation of complex, which have sufficient hydrophobicity [37].” I agree that pH plays a crucial role in many extraction procedures, but why is a reference dedicated to nickel cited in the context of cadmium analysis? In the cited work, APDC is used as a complexing agent, and 1-dodecanol and ethanol are employed as the extraction and dispersive solvents, respectively. Furthermore, this reference discusses a different type of extraction method (dispersive liquid–liquid microextraction based on solidification of floating organic drop).
[37] Wang, Y., Zhang, J., Zhao, B., Du, X., Ma, J., & Li, J. (2011). Development of dispersive liquid–liquid microextraction based on solidification of floating organic drop for the determination of trace nickel. Biological trace element research, 144(1), 1381-1393.
Sentence: “The interaction of metal-ligand for complex formation and efficiency of extraction is directly dependent on pH level of solution.” While this is a statement, it lacks the theoretical explanation of which complex can actually form. Since the formula of PvbXa is depicted in Figure 1, it could be beneficial to propose a formula for the complex (between PvbXa and Cd) and discuss its stability during pH changes. This additional information would indeed be interesting.
Why are you duplicating information, especially from references 40, 41, and 42? This information can be found in both Table 6 and the text. “Shamsipur et al prepared a Natural DES–based ultrasound-vortex-assisted DLLME determination of trace level of Cd ions in food and water. The linearity was found in range of 0.001–7.5 µgL−1 , (R2= 0.995). The planned method provided good LOD 0.37 × 10−4, LOQ 1.24 × 10−4 µgL−1. The PF factor was obtained 125 and RSD% was 2.65%. The 95-99% recovery of Cd was reported by used proposed procedure [40]. Elik and Altunay reported the MIL-DLLME procedure for recovery of Cd ion from different water and food samples. The dynamic range for recovery of Cd(II) was 2–700 ng mL−1. The LOD 0.6 and LOQ was 2.0 ng mL−1. The RSD% was found 1.5% with EF was 172 and 98% recovery of Cd was reported [41]. Yang et al developed graphene oxide from pencil for solid-phase microextraction of Cd by using GF-AAS. Calibration curve for Cd ions was linear range 0.04–0.26 µg L−1 with LOD 0.005 µg L−1. The RSD was 2.1% with EF value was 25. The recovery of Cd in tap water, river, and pond water was found in ranged from 94 to 105% [42].”
Supplementary information
What was the procedure used for interference studies (Table S2)? Initially, the concentration of Cd was 100 µg L-1, followed by the addition of an elevated concentration of a potentially interfering ion. Afterward, was the preconcentration of Cd conducted? A preconcentration factor of 200 was employed, resulting in a determined concentration of Cd of approximately 20,000 µg L-1?
Legends for Figures S1-S9. It would be helpful to include the parameters that remained constant while changing the selected parameter.
Author Response
Author's Reply to the Review Report (Reviewer 4)
Please provide a point-by-point response to the reviewer’s comments and either enter it in the box below or upload it as a Word/PDF file. Please write down "Please see the attachment." in the box if you only upload an attachment. An example can be found here.
* Author's Notes to Reviewer
FileEditViewInsertFormatToolsTableHelp
Paragraph
P
0 WORDS
Word / PDF
or
Formun Altı
Formun Üstü
Review Report Form
Open Review
(x) I would not like to sign my review report
( ) I would like to sign my review report
Quality of English Language
(x) I am not qualified to assess the quality of English in this paper
( ) English very difficult to understand/incomprehensible
( ) Extensive editing of English language required
( ) Moderate editing of English language required
( ) Minor editing of English language required
( ) English language fine. No issues detected
|
Yes |
Can be improved |
Must be improved |
Not applicable |
|
|
Does the introduction provide sufficient background and include all relevant references? |
( ) |
( ) |
(x) |
( ) |
|
Are all the cited references relevant to the research? |
( ) |
(x) |
( ) |
( ) |
|
Is the research design appropriate? |
(x) |
( ) |
( ) |
( ) |
|
Are the methods adequately described? |
( ) |
(x) |
( ) |
( ) |
|
Are the results clearly presented? |
( ) |
(x) |
( ) |
( ) |
|
Are the conclusions supported by the results? |
( ) |
(x) |
( ) |
( ) |
Comments and Suggestions for Authors
A valuable piece of work with numerous interesting results can be observed in your text; nevertheless, I do have some comments and questions.
The abbreviation for cadmium is mentioned twice in the abstract: once in the first line and again in the fourth line. This repetition is unnecessary.
Response: We have removed the repeated abbreviation of cadmium in abstract section as suggested by reviewer.
Why is the term “spectrophotometry” specifically used for FAAS, while “spectroscopy” or “spectrometry” is used for other spectrometric methods? Additionally, other name such as “flame atomic absorption spectrophotometer” for FAAS can be also found on page 2, paragraph 2, line 2.
Response: spectroscopy is a group of physical methods that decompose radiation according to a certain property such as wavelength, energy, mass, etc. The resulting intensity distribution is called spectrum. While spectrophotometry is the quantitative measurement of spectra using a spectrometer. The recording method is called spectrophtography and the recording (graphical representation) itself is called spectrogram".
Sentence: “The FAAS showed great consideration for assess and extraction of metals because of low cost, easy to operate, fast response and best precision and accuracy.” (page 2, paragraph 2) I understand your point, but FAAS doesn't prioritize the thorough extraction of metals. It primarily serves as a quantification method, finding utility in trace analysis when paired with an efficient separation/preconcentration technique for the target analyte.
Response: Thank you for your comments. The FAAS is used for the quantitative analysis of metals or other species and its capable to detect the samples at trace level and mostly used for elemental analysis. The FAAS is very fast technique as compared to other technique.
Reference number 15 is duplicated: once as a numeral and another time as “(Xing et al., 2020)” (page 2, paragraph 1, line 15).
Response: Thank you for your suggestion. We have removed the duplicated reference from the manuscript.
Sentence: “The low detection limit of FAAS and matrix constituents of real samples, it is necessary to use preconcentration techniques, solid phase extraction, dispersive liquid liquid microextraction, etc. [24-26].” Detection limit of FAAS is relatively high. I think you wanted to write: The absence of a low detection limit and the presence of various co-existing matrix components are the two main reasons why employing an effective preconcentration technique becomes necessary for reliable quantification of trace elements using FAAS.
Response: We have re-arraigned the above suggested sentence, hope it will be acceptable.
While the abbreviation of VA-dSPµE is explained in the abstract, there is no explanation provided when it is first used in the main text (page 2, paragraph 4, line 2).
Response: Thank you for your suggestion, we have provided explanation about VA-dSPµE in the main text as suggested.
In the introduction, nearly half of the text is devoted to rice, with no mention of cadmium in other foodstuffs. Don't we have any information about cadmium in other types of food? Your interest wasn't limited to just rice samples; you also examined apple, spinach, salad, tomatoes, onion, oat, corn, aubergine, wheat, and mushroom samples.
Response: Thank you for your suggestion, We have added information about cadmium in other food samples as suggested for reviewer.
I believe that the standout aspect of your work is the utilization of a novel sorbent for cadmium, namely PvbXa. While the introduction merely contains a single sentence about this significant development, it lacks an elaboration on the theory or hypothesis behind the sorption mechanism between PvbXa and Cd. This absence of discussion could be particularly intriguing, especially considering the primary focus of your work.
Response: The following sentence was added into the “Abstract” section:
A new polyvinyl benzyl xanthate (PvbXa) was synthesized and used as a new adsorbent in this work. It contains pendant sulfide groups on the main polystyryl chain. Using this new adsorbent, PvbXa, vortex………..
And added in the “Introduction section” as shown below:
….changing the polymer topology are of great importance in polymer science [29]. Well known that the high reactivity of cadmium (II) to sulphide moieties, we prepared a new adsorbent with high sulphide content. In order to synthesize this new type of adsorbent, poly(vinyl benzyl chloride) was reacted with potassium salt of xanthate. This new polymer is also a a macro RAFT agent and is important for block/graft copolymer synthesis [30].
Part 2.4.
Sentence: “Collected water samples were first filtered using a micro filter into a 250 mL beaker.” What is the size pore of the filter you used? No acid was added to your water samples after collection, right? (just for stabilization)
Response: Collected water samples were filtered using a 0.45 µm membrane filter into a 250 mL beaker and acidified and stored at +4 °C until analysis.
Sentence: “It was then digested via evaporation on the heating plate until the final volume was 10 mL.” From this, I interpreted it as a simple evaporation process. However, does this imply that the water samples were preconcentrated 25 times in this step (from 250 mL to 10 mL)? I couldn't find any information about this preconcentration in the main text.
Response: We did not make evaporation to water samples from 250 mL to 10 mL. This point is corrected. Water samples were applied to Section 2.6. VA-dSPµE procedure.
Sentence: “Collected food samples were first homogenized by pulverizing with a lab grinder.” Did all your actual solid samples undergo the same process? Were apple, spinach, salad, tomatoes, onion, aubergine, and mushroom samples not dried before being subjected to this homogenization process?
Response: Yes, all food samples were first dried and then homogenized with a laboratory blender. To avoid confusion, the phrase "after the collected food samples were dried" was added before the relevant sentence.
Sentence: “Then, a 1:4 mixture of hydrogen peroxide and nitric acid was added to the samples.” What volume of this mixture was used?
Response: Hydrogen peroxide and nitric acid were used as 1 mL and 4 mL, respectively. These volumes have been added to the relevant text.
Part 2.6.
Text: “To get the Cd(II) ions adsorbed on the solid back into the measurement solution, 1250 µL of EtOH was added and vortexed for 120 s. Finally, Cd determination was performed by injecting the death solution into the atomization section of the FAAS.” Does this mean that the sorbent was dissolved? If elution was performed, another separation step should be included in the subsequent procedure after shaking the sorbent with the eluting agent, such as centrifugation or filtration.
Response: No, the sorbent did not dissolve. The goal here is to take the Cd(II) ions adsorbed to the sorbent back into the aqueous solution and make measurements.
Part 2.7.
Sentence: “The 30 experimental run drawn in the factorial design used 4 variables pH, sorbent amount mg, mixing time (min) and sample volume mL Table 2.” For consistency, maybe (mg) and (mL) should also be in parentheses. Maybe something like “it can be seen in Table 2” can be incorporated to this sentence.
Response: In line with your suggestions, the relevant arrangement has been made.
Part 3.2.
Sentence: “Therefore, EtOH volume 1200 µL was adjusted for study.” In section 2.6., a volume of 1250 µL is mentioned. Which value is correct?
Response: We have corrected the suggested values of EtOH, we have adjusted 1250 µL of EtOH in our study.
Part 3.3.
Sentence: “Present VA-dSPµE method has highly selective for uses to water and environmental samples for extraction the Cd ions.” Maybe “food samples” can be also mentioned in this sentence.
Response: We have adjusted the sentence in section 3.3 according to reviewer suggestion, hope it will be acceptable.
Parts 3.4. and 3.5.
In the main text, “µg L-1” is used as the concentration unit, while in tables 3, 4, and 5a, “ng mL-1” is used. Please use the same unit for this parameter throughout the entire article.
Response: Thank you for your comments, We have adjusted the units according to suggestion and changed used ng ml-1 to µg L-1.
While the abbreviation “VA-dSPµE” is used in the text, the abbreviation “VA-SPME” is used in the titles of tables 3, 4, 5a, and 5b. Please use the same abbreviation throughout the entire article.
Response: Thank you for your good comments, we have adjusted the abbreviation and added same abbreviation throughout the entire article and changed VA-SPME to VA-dSPµE in tables 3,4, 5a and 5b.
Text: “Intraday precision (N=5) was found 2.4, 3.1 and 3.6% for 1, 50 and 100 µg L-1 of Cd(II), respectively. Inter-day precision (N=5) was found 2.7, 3.7 and 4.3% for 1, 50 and 100 µg L-1 of Cd(II).” This trend of RSD values is unusual. Typically, when analyzing higher concentrations, lower RSD values can be achieved.
Response: Thank for your suggestion, our research showed RSD is less than 5%, it is acceptable, our research higher RSD was 4.3%, it may be due to the instruments error, samples preparation error or other contamination found in samples.
Sentence: “The PF, EF and ER of VA-dSPµE method was achieve 200, 100, 97.1% respectively (see Table 3).” How were calculated the PF and EF values? Sometimes, published literature can contain confusing information about these parameters. An explanation could help more readers understand the difference between these values.
Response: The EF was calculated by using the ratio of direct calibration curves’s slopes obtained with and without VA-dSPµE method. The PF was calculated from the ratio of initial volume to final volume (200/1.250=160). This point is corrected.
Table 5.a.
I'm having trouble understanding how the spiked samples were prepared and measured. Is the obtained concentration value (for Cd in original sample without the spike) valid for your samples after evaporation (from 250 mL to 10 mL)? Was 75 µg L-1 of Cd added before the extraction procedure? Then, was the sample preconcentrated 200 times? Finally, was a measured concentration of around 15,000 µg L-1 achieved? Is the linear range 0.20 – 150 µg L-1? Additionally, it seems that even Cd in the original sample (without the spike) can be preconcentrated 200 times after using the extraction procedure. I'm quite confused.
Response: We did not make evaporation to water samples from 250 mL to 10 mL. Water samples were applied to Section 2.6. VA-dSPµE procedure. Firstly, we applied to water samples to Section 2.6. VA-dSPµE procedure then we spiked 75 µg L-1 of Cd to the samples after the extraction procedure. Cd in the original sample (without the spike) can be preconcentrated 160 times after using the extraction procedure. We used spike (standard additional method) for accuracy of the method. In high concentrations, we made dilution of the samples.
Part. 4.
Sentence: “The value of pH is a critical factor that may significantly affect the efficiency of extraction and metal separation through microextraction procedure comprises prior formation of complex, which have sufficient hydrophobicity [37].” I agree that pH plays a crucial role in many extraction procedures, but why is a reference dedicated to nickel cited in the context of cadmium analysis? In the cited work, APDC is used as a complexing agent, and 1-dodecanol and ethanol are employed as the extraction and dispersive solvents, respectively. Furthermore, this reference discusses a different type of extraction method (dispersive liquid–liquid microextraction based on solidification of floating organic drop).
[37] Wang, Y., Zhang, J., Zhao, B., Du, X., Ma, J., & Li, J. (2011). Development of dispersive liquid–liquid microextraction based on solidification of floating organic drop for the determination of trace nickel. Biological trace element research, 144(1), 1381-1393.
Response: Thank you for your suggestion, we have discussed general effect of pH on the microextraction which is not specific for cadmium. We think nickel is also toxic heavy metal like Cd and its properties are also similar to Cd.
Sentence: “The interaction of metal-ligand for complex formation and efficiency of extraction is directly dependent on pH level of solution.” While this is a statement, it lacks the theoretical explanation of which complex can actually form. Since the formula of PvbXa is depicted in Figure 1, it could be beneficial to propose a formula for the complex (between PvbXa and Cd) and discuss its stability during pH changes. This additional information would indeed be interesting.
Response: And added in the “Introduction section” as shown below:
….changing the polymer topology are of great importance in polymer science [29]. Well known that the high reactivity of cadmium (II) to sulphide moieties, we prepared a new adsorbent with high sulphide content. In order to synthesize this new type of adsorbent, poly(vinyl benzyl chloride) was reacted with potassium salt of xanthate. This new polymer is also a macro RAFT agent and is important for block/graft copolymer synthesis [30].
The suggested complex formation can be designed as shown in Figure SI-1.
Figure SI-1. The suggested complex formation of Cd (II) with sulphide groups of PvbXa.
Why are you duplicating information, especially from references 40, 41, and 42? This information can be found in both Table 6 and the text. “Shamsipur et al prepared a Natural DES–based ultrasound-vortex-assisted DLLME determination of trace level of Cd ions in food and water. The linearity was found in range of 0.001–7.5 µgL−1 , (R2= 0.995). The planned method provided good LOD 0.37 × 10−4, LOQ 1.24 × 10−4 µgL−1. The PF factor was obtained 125 and RSD% was 2.65%. The 95-99% recovery of Cd was reported by used proposed procedure [40]. Elik and Altunay reported the MIL-DLLME procedure for recovery of Cd ion from different water and food samples. The dynamic range for recovery of Cd(II) was 2–700 ng mL−1. The LOD 0.6 and LOQ was 2.0 ng mL−1. The RSD% was found 1.5% with EF was 172 and 98% recovery of Cd was reported [41]. Yang et al developed graphene oxide from pencil for solid-phase microextraction of Cd by using GF-AAS. Calibration curve for Cd ions was linear range 0.04–0.26 µg L−1 with LOD 0.005 µg L−1. The RSD was 2.1% with EF value was 25. The recovery of Cd in tap water, river, and pond water was found in ranged from 94 to 105% [42].”
Response: Thank you for your kind suggestion, We have removed information from table 6 and such as reference 40, 41 and 42.
Supplementary information
What was the procedure used for interference studies (Table S2)? Initially, the concentration of Cd was 100 µg L-1, followed by the addition of an elevated concentration of a potentially interfering ion. Afterward, was the preconcentration of Cd conducted? A preconcentration factor of 200 was employed, resulting in a determined concentration of Cd of approximately 20,000 µg L-1?
Response: The experimental steps were performed as you described. Here, the concentration of the added Cd(II) ions was written incorrectly. ug L-1 is written instead of ng mL-1. In fact, 100 ng L-1 Cd(II) was added. After enrichment, the maximum amount can be 10 ug L-1.
Legends for Figures S1-S9. It would be helpful to include the parameters that remained constant while changing the selected parameter.
Response: It was corrected.
Submission Date
10 August 2023
Date of this review
30 Aug 2023 14:38:21
Formun Altı
Formun Altı
Round 2
Reviewer 3 Report
Although a motivation to justify the aligement of this paper to the scope of this journal was expected, based on the other reviewers suggestions I will report that the manuscript is suitable.
Reviewer 4 Report
The changes you made to the manuscript are acceptable to me. I recommend publishing your article.